# Recent Advances in the Design of Topical Ophthalmic Delivery Systems in the Treatment of Ocular Surface Inflammation and Their Biopharmaceutical Evaluation

**DOI:** 10.3390/pharmaceutics12060570

**Published:** 2020-06-19

**Authors:** Roseline Mazet, Josias B. G. Yaméogo, Denis Wouessidjewe, Luc Choisnard, Annabelle Gèze

**Affiliations:** 1DPM, UMR CNRS 5063, ICMG FR 2607, Faculty of Pharmacy, University of Grenoble Alpes, 38400 St Martin d’Hères, France; rmazet@chu-grenoble.fr (R.M.); denis.wouessi@univ-grenoble-alpes.fr (D.W.); luc.choisnard@univ-grenoble-alpes.fr (L.C.); 2Grenoble University Hospital, 38043 Grenoble, France; 3UFR/SDS, University Joseph Ki-Zerbo, Ouagadougou 03 BP 7021, Burkina Faso; josiasbyg@yahoo.fr

**Keywords:** SAID, NSAID, immunosuppressant drugs, topical ophthalmic formulation, recent advances, biopharmaceutical evaluation

## Abstract

Ocular inflammation is one of the most common symptom of eye disorders and diseases. The therapeutic management of this inflammation must be rapid and effective in order to avoid deleterious effects for the eye and the vision. Steroidal (SAID) and non-steroidal (NSAID) anti-inflammatory drugs and immunosuppressive agents have been shown to be effective in treating inflammation of the ocular surface of the eye by topical administration. However, it is well established that the anatomical and physiological ocular barriers are limiting factors for drug penetration. In addition, such drugs are generally characterized by a very low aqueous solubility, resulting in low bioavailability as only 1% to 5% of the applied drug permeates the cornea. The present review gives an updated insight on the conventional formulations used in the treatment of ocular inflammation, i.e., ointments, eye drops, solutions, suspensions, gels, and emulsions, based on the commercial products available on the US, European, and French markets. Additionally, sophisticated formulations and innovative ocular drug delivery systems will be discussed. Promising results are presented with micro- and nanoparticulated systems, or combined strategies with polymers and colloidal systems, which offer a synergy in bioavailability and sustained release. Finally, different tools allowing the physical characterization of all these delivery systems, as well as in vitro, ex vivo, and in vivo evaluations, will be considered with regards to the safety, the tolerance, and the efficiency of the drug products.

## 1. Introduction

Ocular inflammation is considered as a major eye disorder and many reports demonstrated that topical administration of anti-inflammatory drugs, non-steroidal (NSAIDs) [1] and steroidal (SAIDs) [2], are effective in treating ocular surface and anterior segment inflammation, including pain and post-operative inflammation, seasonal allergic conjunctivitis [3,4], and age-related macular degeneration [5]. Furthermore, some immunosuppressive agents, such as ciclosporin A (CsA), demonstrated their efficiency in the treatment of keratitis associated with dry eye disease (DED) [6,7]. The major challenge in the therapeutic management of ocular inflammation is rapid treatment in order to reduce the risk of visual impairment while limiting side effects. Topical administration is the most preferred route for the management of ocular inflammations as it is (i) easy to handle, (ii) non-invasive, (iii) rather well-tolerated [1], and (iv) it provides sufficient ocular drug concentrations, while avoiding the systemic side effects associated with oral administration.

Nevertheless, the ocular drug bioavailability in conventional topical formulations is notoriously poor, as only 1–5% of drug applied to the surface penetrates the cornea. This is the consequence of various protective mechanisms and multiple barriers to drug entry, such as fast nasolacrymal drainage due to high tear fluid turnover and lid blinking, the corneal structure with a hydrophilic stroma sandwiched between the lipophilic epithelium and endothelium, epithelial drug transport barriers, the efflux pump, and the clearance from the vasculature in the conjunctiva [8,9]. Besides these ocular anatomical and physiological constraints, another limiting factor encountered with anti-inflammatory drugs or immunosuppressive agents is their poor water solubility [10,11,12]. Thus, they require complex formula adapted to regulatory specifications, due to the eye fragility, their low ocular bioavailability, and their poor water solubility. As a consequence, a limited number of drugs are marketed as well as a few drug associations with anti-infective molecules [13,14].

Despite these drawbacks, many strategies have been investigated in order to improve their ocular topical bioavailability, such as physicochemical modifications of the active pharmaceutical ingredient (API) in order to favor their absorption or the development of formulations ensuring a prolonged corneal residence time of the drug product. 

Concerning the physicochemical modifications of drug molecules, one approach is based on the synthesis of new APIs from the chemical structures of well-known available anti-inflammatory drugs. For instance, new drug molecules were synthesized from propionic acid derivatives of NSAIDs, such as pranoprofen, pyranoprofen, and suprofen [1]. Other molecules were derived from SAIDs, such as clobetasone butyrate, difluprednate, and loteprednol. Unfortunately, these new synthesized molecules did not lead to expected enhanced ocular penetration [1], are more irritating in nature, or have an increased higher risk of side effects [15]. The prodrug approach is another chemical way to enhance drug permeability. Indeed the synthesized inactive prodrug exhibits a better corneal penetration and once in situ, is either chemically and/or enzymatically metabolized to become active [16]. As an example, nepafenac, an amide prodrug of amfenac, belongs to the pharmacological NSAID class of arylacetic derivatives and is commercially available. In vitro nepafenac demonstrated a nearly six-fold greater permeation coefficient than diclofenac [17]. In vivo, nepafenac easily crosses corneal and retinal tissues following topical ocular administration. Thereafter, nepanefac is hydrolyzed to amfenac, which shows high anti-inflammatory properties when used to treat pain and inflammation associated with cataract surgery [18]. Several lipophilic esters of dexamethasone were developed and evaluated for permeability and bioreversion across the rabbit cornea and bovine conjunctival epithelial cells (BCECs). The permeability of phosphate and metasulfobenzoate esters of dexamethasone were restricted across BCECs due to their hydrophilic and ionic character. On the contrary, prodrugs, including acetate, propionate, and butyrate esters, demonstrated better permeability, which increased with the ester lipophilicity. The valerate ester conjugate, being highly lipophilic, easily crosses the corneal epithelium while the hydrophilic stroma acts as a barrier and allows a depot of lipophilic prodrug until hydrolysis to the parent dexamethasone. The hydrolysis of valerate ester is very slow in the cornea, suggesting, for this prodrug, possible use as a sustained drug release system. Lallemand et al. [19,20] developed a series of amphiphilic acidic prodrug molecules having an approximately 25,000 times higher solubility than ciclosporin (CsA) in isotonic phosphate buffer solution at pH 7. These prodrugs are quantitatively hydrolyzed in artificial tears to release CsA within 1 min. Prodrug conversion into the parent molecules was significantly faster in tear fluid than in a buffer at physiological pH, indicating that the hydrolysis is enzyme mediated. Aqueous formulations of these esterified CsA prodrugs were well tolerated and have shown a significant improvement in basal tear production in dry eye disease (DED) [12,21]. Aqueous prodrug solutions have also been evaluated for their efficacy in the treatment of corneal graft rejection and it was found that prodrug eye drops applied five times a day were therapeutically equivalent to a 10 mg/kg/day intramuscular injection in rats [22]. Recent studies have shown that 2% prodrug solutions have a 200- to 500-fold higher conjunctival permeability than the conventional 2% CsA in oil formulation. An accumulation of CsA prodrug formulations in the cornea to form large tissue deposits that provide a sustained release effect over prolonged periods of time was also observed. However, prodrug formulations did not show much improvement in the permeability across the cornea and into the aqueous humor compared with the conventional CsA emulsion, probably because of the rapid conversion of the prodrug into CsA at the corneal surface. This depot formation could have the added advantage of overall poor systemic absorption of prodrug formulations, reducing the incidence of systemic complications and immunosuppression. Prodrug formulations with CsA concentrations higher than 2% are currently under investigation for safety and toxicity [23,24,25].

The prodrug approach is tailor-made to improve solubility, stability, or permeability characteristics to lead molecules without causing any damage to the biological barriers involved. Despite increased research work, there are only a few prodrug products due to their poor stability in the aqueous environment [26]. 

Ocular retention of the drug product combined or not with corneal penetration enhancers can also improve drug bioavailability. These approaches are carried out through conventional formulations, i.e., eye drop solution or suspension, ointments, and hydrogels, for example, by using mucoadhesive agents in these formulations. Furthermore, other sophisticated drug delivery systems have been achieved, such as liposomes, micro-polymer systems, or solid inserts. Iontophoresis is a non-invasive technique, applied with ionized active ingredient for anterior and posterior ocular disorders. It can achieve higher bioavailability and reduce clearance as compared to topical eye drops [27]. In parallel with these novel drug delivery systems, researchers have focused on the development of new functional materials as well as innovative formulations based on the use of combined strategies. Finally, the ideal drug delivery system should administer accurate and therapeutic concentrations of the drug over a specified time, correlated with the ophthalmic affection disorder. It should also be easy to handle and manufacture, and should remain stable over the whole ocular surface, be biocompatible, preferably be biodegradable, and be free of toxic side effects. 

Various reviews have been published in this area, covering the administration of anti-inflammatory drugs based on NSAIDs [1] and SAIDs [28] to the anterior and posterior segments of the eye, respectively. The reviews of Janagam et al. [29], Lalu et al. [30], and Cholkar et al. [31] focused on the development of novel nanosystems for drugs from various pharmacological classes. The present review gives an updated insight into topical ophthalmic administration of SAIDs, NSAIDs, and immunosuppressive agents in order to control ocular inflammation. Indeed, immunosuppressive agents are specifically used in the treatment of inflammation associated with dry eye syndrome. Additionally, the review provides exhaustive information concerning the marketed specialties of the French, European, and US markets, brands, and generics, specifying their indications and their complete formulation. In addition, conventional formulations and innovative ocular drug delivery systems are discussed. The different characterization tools for biopharmaceutical evaluations of the systems are also considered.

## 2. NSAIDs, SAIDs, and Immunosuppressive Agents

### 2.1. Chemical Family

Corticosteroids have a C21 structure, presenting a steroid nucleus derived from cholesterol [32]. From this backbone, numerous drugs vary from differing functional groups and oxidation states [33]. Topical corticosteroids used in ophthalmology can be classified as ketone or ester steroids. Loteprednol is the only ester steroid drug presenting an ester instead of a ketone group at the C20-position, responsible for the cataractogenic side effect [34,35].

Unlike corticoids, NSAIDs do not include a steroid nucleus and are a heterogeneous group of compounds of different chemical classes.

As shown in Table 1, the ophthalmic topical route is largely under-endowed in anti-inflammatory drug specialties compared to other routes of administration. Table 1 summarizes the anti-inflammatory drug molecules commercially available for oral, parenteral, and topical ophthalmic administrations in France, the EU, and the USA, as well as their chemical class [1,36,37,38,39,40]. On February 26, 2019, a total of 40 NSAIDs or SAIDs were marketed for oral, parenteral, or topical ocular administrations, only 14 (35%) of which concerned the topical ocular route. Among these 14 drugs, 5 of them (12.5%) are actually available only for the topical ophthalmic route. Those are bromfenac, difluprednate, fluorometholone, loteprednol etabonate, and nepafenac.

Immunosuppressant agent cannot be classified according to their chemical family. In the present review, they are further classified according to their mechanism of action. The only immunosuppressive agent marketed in Europe, the USA, and France for topical ocular administration is ciclosporin. Note that a specialty based on tacrolimus 1 mg/mL TALYMUS^®^ is available in France as compassionate use, and a French specificity called ‘ATUn’ with nominative temporary use authorization and is marketed only in Japan.

Table 2 includes all the brand name products of NSAIDs, SAIDs, or CsA marketed for the ophthalmic topical route and used in the USA, the EU, and France as of February 26, 2019, except the generic specialties. The combinations of anti-inflammatory drugs with other pharmacological classes of molecules are also listed.

### 2.2. Mechanism of Action

Inflammation corresponds to a set of mechanisms of defense, physiological and pathological, by which the organism recognizes, destroys, and eliminates all the substances foreign to it. It is a dynamic process with several successive steps in which the membrane phospholipids will be degraded in arachidonic acid by phospholipase A2, resulting in the release of pro-inflammatory mediators, including prostaglandins, thromboxanes, leukotrienes, and eicosanoids. The corticosteroids and NSAIDs both inhibit prostaglandin formation, but their pharmacological properties differ by their place of action in the inflammatory cascade (Figure 1).

The corticosteroid agents inhibit the arachidonic acid pathway indirectly through the induction of lipocortin synthesis, which inhibits the phospholipase A2 enzyme, therefore preventing the production of all proinflammatory mediators, including the arachidonic acid cited above [1,35,41]. Despite their chemical heterogeneity, NSAIDs share similar therapeutic properties. They act solely on the action of cyclooxygenase (COX), inhibiting among others the formation of prostaglandins [1,42,43]. Conventional NSAID agents inhibit both COX-1 and COX-2 in a nonselective way.

Concerning the immunosuppressive agents, their design is based on the control of the exacerbated immune response. The pathophysiological means of this concept is to modulate the action of mononuclear cells, with T cells being the main targets. Immunosuppressive agents have different molecular targets, and an important drawback in their use is that they also inhibit the normal immune system response. Depending on their mode of action, immunosuppressive drugs can be classified in three different groups: Inhibitors of the calcineurin pathway, cytototoxic or antiproliferative drugs, and specific antibodies [44]. Actually, CsA is the only immunosuppressive agent used for the ophthalmic route of administration in France, the EU, and the USA. CsA is an inhibitor of the calcineurin pathway and mainly acts by inhibition of T cells by blocking cytokines’ transcription genes, like Interleukine 2 and Interleukine 4 [45], and stimulates the autoinhibitory action of calcineurin A, which results in a reduction of phosphatase activity, thus causing inflammation [46,47]. Furthermore ciclosporin blocks both the p38 and c-Jun *N*-terminal kinase (JNK) pathways in addition to calcineurin-blocking activity [48]. JNK and p38 work in the stress response like inflammation and apoptosis [49,50,51].

### 2.3. Sites of Action/Therapeutic Uses

Topical SAIDs are widely prescribed as anti-allergic or anti-inflammatory drugs for the anterior segment of the eye (Figure 2), combined or not with anti-infectious drugs (Table 3). In order to treat conjunctival diseases, SAIDs can be used to treat allergic conjunctivitis, blepharoconjunctivitis, and corneo-conjunctival burns. Regarding corneal diseases, the indications are the treatment of immune and bacterial keratitis, in any case herpetic or mycotic. The anti-inflammatory effect is highly used in post-operative inflammation, such as cataract or glaucoma surgery, or in the prevention of corneal graft rejection, as an immunosuppressive agent [52,53].

Topical ophthalmic NSAIDs are sometimes indicated but are less prescribed to treat post-operative inflammation, e.g., following cataract surgery. They have also shown benefits by preventing intraoperative miosis, improving treatment of seasonal allergic conjunctivitis, and reducing post-operative pain [1,3].

### 2.4. Side Effects

There are many important ocular side effects of NSAIDS, SAIDS, and immunosuppressive agents. Topical administration of NSAIDs is common, but this treatment has clinically significant side effects, including ulceration and corneal perforation [56]. The adverse effects associated with the use of corticosteroid eye drops are different. These include elevated intraocular pressure and induced glaucoma, cataract formation, delayed wound healing, and increased susceptibility to infection [57]. Furthermore, the most common reported side effect of CsA is ocular burning, reported in 17% of patients, and approximately 3% of patients stop the medication as a result of this side effect [7].

## 3. Formulation for Topical Ophthalmic Drug Delivery Systems

### 3.1. Conventional Formulation

Most conventional ophthalmic dosage forms include ointment, solutions, emulsions, and suspensions, which together account for nearly 90% of the currently available formulations in the United States and Europe. It is usual that water-soluble drugs are delivered through topical instillation in an aqueous solution and water-insoluble drugs are administered topically as ointments or aqueous suspensions [58]. Among the topical dosage forms for ophthalmic drug delivery, eye drop solutions are quite popular since they are relatively well tolerated by patients, and simple to prepare, filter, and sterilize. On February 26, 2019, we identified 93 commercial drugs, brands, and generics, on the USA, European, and French markets. Among these specialties, 35 contain an NSAID as the API, 23 contain SAID, 30 correspond to an anti-inflammatory API associated with anti-infective drugs (1 association with NSAID and 29 associations with SAID), and 5 contain CsA (Figure 3). The marketed medicines are reported in Table 4, Table 5, Table 6 and Table 7. It should be noted that the first line of inactive ingredients corresponds to the preservatives present in the formulation. The composition of some marketed formulations is unfortunately not currently available.

Among these 93 topical ocular specialties, 42 are formulated as solutions, 28 as suspensions, 15 as ointments, 5 as emulsions, 2 as gels, and one as an intracanalicular insert (Figure 4).

#### 3.1.1. Ointments

The ophthalmic ointment base is generally made of mineral oil and petrolatum. Due to their composition, they present the great advantage of increasing the contact time of the drug (two to four times longer). The ointment bases are generally either monophasic bases in which the vehicle forms one continuous phase, or biphasic systems, in which an emulsion of oil and water is created. The ointments may cause discomfort to patients. They blur the vision due to the refractive index difference between the tears and the non-aqueous nature of the ointment and inaccurate dosing [59,60]. Consequently, they are less marketed, with only 15 specialties being counted among the 93 products listed in the Table 4, Table 5, Table 6 and Table 7.

#### 3.1.2. Eye Drops

##### Solutions

Most of the topical ophthalmic preparations available today are in the form of aqueous solutions. A homogeneous solution dosage form offers many advantages, including the simplicity of large-scale manufacturing, easy handling, and good tolerability. The factors that must be taken into account while formulating aqueous solution include the selection of the appropriate salt of the drug substance to achieve the therapeutic concentration required. The compatibility of other formulation components, such as the preservative or buffer salts, should be considered as well as the inertia of the primary packaging. Some typical physical parameters, including the pH, osmolality, viscosity, color, and appearance of the product, must be suitable for ocular administration. Usually, aqueous solutions are easily manufactured by the dissolution of active and inactive compounds before sterilization by filtration or autoclaving. Nevertheless, most of the recently developed drugs are hydrophobic and have limited solubility in water [60]. in total, 41 anti-inflammatory specialties and one immunosuppressive specialty among the 93 listed in Table 4, Table 5, Table 6 and Table 7 are formulated as ophthalmic solutions.

##### Suspensions

Ophthalmic suspensions may be defined as a fine dispersion of insoluble API in water, which is considered as the most suitable solvent for ocular administration. Eye drop suspensions appear to be an unavoidable alternative to formulate some interesting API, which are hydrophobic and then have limited solution in water. What is expected with administered suspensions is that solid drug particles will be retained in the conjunctival cul-de-sac and will then improve the dug contact time compared to an eye drop solution. Solid drug particles dissolve progressively, leading to improved bioavailability [61]. However, it must be emphasized that the formulation of eye drop suspensions is a real challenge. One of the main parameters to take into account is the size of the solid API suspended. For reasons of patient comfort, the average particle size in most eye drop suspensions is below 10 µm [62]. Likewise, the morphology of solid particles, i.e., irregular shape and crystallinity, must be considered with regards to irritation of the ocular mucosa. Regarding the tolerated particle size, due to the larger surface area deployed, smaller-sized drug particles dissolve more or less quickly in the precorneal pocket liquid and the drug is absorbed into ocular tissues while the larger particles dissolve more slowly, prolonging the contact time and the availability of the drug.

Another concern of the formulations is the addition of adequate inactive ingredients for many beneficial reasons, i.e., preservative to prevent microbiological contamination, suspending agents to limit rapid particle settling or caking, and surfactants used as wetting or stabilizing agents. In some formulations, hydrophilic cyclodextrins (CDs) have been added as complexing agents for solubilizing hydrophobic drug molecules. The CD may also act as absorption promoters [62]. Finally, the redispersibility of drug particles by shaking the container must be effective to ensure the mean dose and the uniformity of amounts administered under therapeutic conditions. In addition to the complexity of the formulations, the technological aspects of the manufacture of suspensions are also to be considered. Indeed, the fabrication of these dosage forms is unconventional and requires specific equipment, such as suspension aseptic ball milling. The sterile product is subsequently filled into sterile containers, which are hermetically sealed under an aseptic environment, i.e., class A grade.

Despite all the difficulties encountered in the formulation and manufacture of eye drop suspensions, some very interesting pharmaceutical products have already reached the market. To our knowledge, almost 27 suspensions are marketed in Europe and in the USA. Most of these specialties are from the 20th century, but some of them are relatively recent, showing the interest in these ophthalmic dosage forms (Table 3, Table 4 and Table 5). One representative example is NEVANAC^®^, which was launched in the USA market in 2007 for the treatment of post-operative inflammation after cataract surgery. NEVANAC^®^ is a 0.1% suspension of a nepafenac, which is described chemically as 2-*amino*-3-benzoylbenzeneacetamide. This API is an amide lipophilic prodrug, which is expected to be deaminated by hydrolytic enzymes in aqueous humor to amfenac (2-*amino*-3-benzoylbenzeneacetic acid), known as an NSAID, which has unique time-dependent inhibitory properties for COX-1 and COX-2. The prodrug nepafenac is less polar or nonionized and offers better corneal penetration [63]. Note that a new suspension formulation of nepafenac 0.3% has already been developed by Novartis in the USA (ILEVRO^®^) but is not commercialized in the European Union (Table 4).

Another example is the eye drop suspension TOBRADEX^®^, which is a combination product of two APIs, an antibiotic, tobramycin (0.3%), and a steroid, dexamethasone (0.1%). This commercial product represents one of the widely used steroids, indicated when superficial bacterial ocular infection or a risk of bacterial ocular infection exist. It is interesting to note that TOBRADEX^®^, which came to the market in 1997, has continued to be improved recently. Indeed, a new formulation was developed and launched as TOBRADEX^®^ST by Alcon Laboratories, Inc., with the scope to increase the pharmacokinetic characteristics as well as patient compliance compared to TOBRADEX^®^. The combination of the API in the TOBRADEX^®^ST was tobramycin 0.3% and dexamethasone 0.05%, which is half of the TOBRADEX^®^ content. Concerning the formulation, the main change was the replacement of the suspending agent Hyetellose (hydroxyethylcellulose) present in TOBRADEX^®^ by xanthan gum in TOBRADEX^®^ST. The consequences of these modifications were that the anti-inflammatory and anti-infective activities were improved by the new suspension formulation. The explanations could be that xanthan gum, which is an anionic polysaccharide with a repeating unit of two d-glucose, two d-mannose, and one d-glucuronic acid residues, forms an ionic interaction with tobramycin to decrease the viscosity of the suspension. This interaction reduces the sedimentation of dexamethasone and improves the suspension characteristics. After the eye drop is instilled, the pH 7 and ionic content of tears disrupts the interactions between xanthan gum and tobramycin, leading to an enhanced viscosity of the eye drop, which increase its ocular retention and then improves the bioavailability of the drugs. Despite these substantial differences, TOBRADEX^®^ST appears to be clinically equivalent to the older formulation [64]. Through these two examples, it is possible to say that an eye drop suspension may be of particular interest for the ocular formulation of some APIs.

#### 3.1.3. Gels

Gels are intended to be introduced into the conjunctival cul-de-sac or to be applied to the conjunctiva. These semi-solid pharmaceutical presentations are made of polymers, presenting the ability to swell in aqueous solvents, which makes it possible to increase the contact time of the preparation, reduce the elimination rate, and obtain a prolonged release of the active ingredient [65]. They reduce the frequency of administration and side effects and consequently improve compliance. They have formed a popular strategy in the early research stages of ocular drug delivery. Hydrophilic gels (hydrogels) mainly have the advantage of being transparent and therefore less disturbing to the vision than ointments. However, the drying of the preparation over time and especially at night leads to the formation of deposits that are often not well accepted by the patient. For this reason, it is preferable to use gels during the day rather than at night.

The main inactive ingredients (excipients) used are viscosity modifiers, which slightly increase the viscosity of the product. As previously described, the latter can also be used to stabilize suspensions or as a substitute for tears (artificial tears). These polymers form transparent gels, are sterilizable and water miscible, and have rheological properties that are adapted to be easily spread on the surface of the eye [65]. A distinction is made between preformed gels (already in the form of gels at the time of application) and in situ gels, applied as a solution, whose gelling mechanism takes place after instillation, due to physicochemical change inherent to the ocular environment (variation in the pH, temperature, or ions). Among the most common polymers used to obtain preformed gels are cellulosic derivatives (hydroxypropylmethylcellulose (hypromellose), methylcellulose, hydroxyethylcellulose, carboxymethycellulose), polyvinylalcohol (PVA), carbomers, and hyaluronic acid. Sometimes, a combination of polymers is possible. In situ gels are instilled in liquid form, like a simple eye drop, allowing accurate and precise administration. They provide good sustained release properties. Once- or twice-a-day dosing is the typical expectation of these gel systems. For example, polymers, such as gellan gum and sodium alginate, are able to form gels in the presence of mono or divalent cations while poloxamer rare temperature-responsive polymers [66].

The anti-inflammatory eye gels on the market fall into the category of preformed gels, with the example of LOTEMAX^®^ (loteprednol etabonate) containing polycarbophil (cross-linked polyacrylic acid) as the viscosifying agent (Table 4). One can note that some pharmaceutical compositions contain hydrophilic polymer agents (polyvinyl alcohol, carboxymethylcellulose, hydroxypropylmethylcellulose) so that they are less fluid, without being classified as gels in the summary of product characteristics. OCUFEN^®^ 0.03% flurbiprofen sodium), ACUVAIL^®^ 0.45% (ketorolac tromethamine), and PREDNISOLONE SODIUM PHOSPHATE EQ 0.9% are some examples listed in Table 4 and Table 5, and are classified as eye drops and aqueous solutions. The effect of viscosity enhancers on drug bioavailability is minimal in humans and their clinical significance is modest [59]. Today, they continue to be used in the formulations of ophthalmic products, but their function is more for patient comfort and/or reasons of bioadhesion rather than viscosity enhancement [58]. Marketed SAID/NSAID-based ocular gels are either presented in preservative-containing multi-dose or single dose packaging. An effort is being made to develop multi-dose packaging free of preservatives.

#### 3.1.4. Emulsions

Emulsions are systems composed of liquid droplets of a liquid A dispersed in another liquid B along with surfactants. Two types of emulsion are described: Water-in-oil and oil-in-water emulsion. These systems are useful, particularly oil-in-water emulsion, in the delivery of poor water-soluble drugs. By keeping the drug in solution, the issue of potential absorption because of the slow dissolution of solid drug particles is avoided. In addition, the blurred vision caused by oils is minimized by the water in the external phase. Furthermore, the concentration of the drug in the oil phase can be adjusted to maximize the thermodynamic activity, thus enhancing drug penetration and bioavailability [67].

DUREZOL^®^ is a topical corticosteroid that is indicated for the treatment of inflammation and pain associated with ocular surgery. The product, approved by the US FDA in June 2008, is a sterile preserved ophthalmic oil-in-water emulsion. It contains a nonionic emulsifying surfactant polysorbate 80 (4%, w/v), sorbic acid as the preservative and castor oil (5%, w/v) as the oily vehicle. Emulsion eye drops offer advantages over suspensions of solubilizing hydrophobic drug in the oily emulsion vehicle, providing uniform doses without the need for shaking before use. William Stringer and Roy Bryant studied the dose uniformity of DUREZOL^®^ emulsion 0.05% versus branded prednisolone acetate ophthalmic suspension 1% (PRED FORTE^®^) and it generic under different simulated patient usage conditions. All the results of their study showed that the dose uniformity of DUREZOL^®^ emulsion was predictable, within 15% of the declared concentration, whereas the drop concentration of PRED FORTE^®^ and generic prednisolone acetate ophthalmic suspensions were highly variable throughout the study depending on whether the bottle of eye suspension was stored upright or inverted as well as the shaking or not of the bottle before use [68]. Furthermore, regarding the in vivo corneal penetration of difluprednate, Yamaguchi et al. found that within 30 min of instillation, the emulsion achieves a concentration 7.4 times higher compared to the suspension. Adfditionally, after 1 hour of instillation, the emulsion led to a higher difluprednate concentration (5.7-fold) in aqueous humor compared to the suspension [69].

RESTASIS^®^ is a 0.05% cyclosporine ophthalmic emulsion approved by the US FDA in 2003. The inactive ingredients are glycerin (2.2%), castor oil (1.25%), polysorbate 80 (1%), carbomer copolymer type A (0.05%), purified water (to 100%), and sodium hydroxide for pH adjustment [70]. Recently, in 2016, the US FDA again approved a new presentation of ciclosporin emulsion as RESTASIS^®^ MULTIDOSE, which has the same composition as RESTASIS^®^ but benefits from the new packaging of a multi-dose bottle with a patented unidirectional valve and air filter technology that eliminates the need to use a preservative. As previously discussed by Ding et al., oil-in-water emulsions are particularly useful in the delivery of water-insoluble drugs that are solubilized in the internal oil phase [58]. By choosing appropriate inactive ingredients, i.e., a new type of emulsifiers or polymeric emulsifiers that are safe and non-irritating, novel ophthalmic emulsion formulations could be achieved with good stability and improved drug bioavailability.

However, castor oil is the inactive ingredient commonly used as the lipophilic phase of the emulsions, although some cytotoxicity towards conjunctival cells has been observed. Indeed, Said et al. showed in an in vitro study that incubating human conjunctival cells with a castor oil vehicle during a 15 min-period of time induced significant cell death. The authors think that this in vitro cytotoxicity could explain the side effects observed in some patients and suggest choosing other lipophilic vector to replace castor oil in emulsion-based ophthalmic formulations [71].

Two other emulsion eye drops contain ciclosporin. IKERVIS^®^ and VERKAZIA^®^ were approved throughout the EU in 2015 and 2018, respectively. They contain ciclosporin at a concentration of 1 mg/mL (0.1%) and are presented as a 0.3-mL single dose. The inactive ingredients are as follows. Medium-chain triglyceride (TCM), a fully saturated triglyceride, was chosen as the oily vehicle instead of castor oil or soybean oil. TCM easily solubilizes ciclosporin, it has a very low viscosity, and is expected to easily spread on the surface of the eye. Tyloxapol (hydrophilic–lipophilic balance (HLB) 12.5) and poloxamer 188 (HLB 29) are two complementary surfactants that ensure the physical stability of the dispersed oil phase within the aqueous phase represented by water for injections. Glycerol could act as a tonicity agent, and sodium hydroxide as the pH adjuster. A cationic surfactant, cetalkonium chloride, is used in the product, thus leading to a cationic emulsion [72]. As reported in the literature, the oil-in-water emulsion has the advantages of optimizing spreading and increasing the residence time on the surface of the eye after instillation of the product. Indeed, electrostatic interactions are created between the positive charges of the emulsion and with the negatively charged mucins present at the ocular surface [73,74]. It therefore appears that IKERVIS^®^ will present a formulation for effective delivery of ciclosporin to the cornea [75].

#### 3.1.5. Use of Penetration Enhancers

The use of an absorption enhancer transiently increases the drug permeability across ocular membranes by decreasing barrier resistance. The surfactants alter the physical properties of cell membranes, by disrupting the tear film and mucin layer as well as the epithelia by loosening tight junctions or by modifying the cell membrane. The benzalkonium chloride (BAK), which is commonly used in formulations for ocular drugs as a preservative, is a cationic surfactant. BAK is known as an irritant even when used at low concentrations (<0.01%). It destabilizes the tear film, thus removing its protection properties. Additionally, it acts on the phospholipid bilayer of cell membranes, inducing morphological changes in the epithelium. Concerning EDTA, also found in ocular formulations as a chelating agent, it is able to disrupt the tight junctions via the extraction of Ca^2+^ [76,77]. Therefore, formulations have changed over the past decade by removing preservatives, such as BAK, and adapting multi-dose primary packaging, for example: COMOD^®^ or ABAK^®^, or by developing single-dose forms. As well, the cyclodextrins (α, β, and γ), which are cyclic oligosaccharides, are able to extract cholesterol and lipids from ocular membranes [77,78]. Finally, crown ethers, bile acids, bile salts (deoxycholate, glycocholate, taurodeoxycholate), and cell-penetrating peptides (TAT, penetration, poly(arginine), poly(serine)) have been the subject of research that shows their role as penetration enhancers. Nevertheless, none of these are currently used in ocular medicines [77]. However, the safety of these enhancers has to be proven before clinical trials and particularly considering the long-term exposure of the ocular tissues to enhancers.

### 3.2. Original Formulations

From conventional formulations, new formulations have been developed to increase the residence time, decrease the frequency of instillation, and finally increase the bioavailability of ophthalmic dosage forms [79]. These formulations innovate by the materials used, by the use of micro or nanaoparticles, or by the use of combined strategies.

#### 3.2.1. Contact Lens

Contact lenses are curve-shaped discs prepared from polymeric materials originally designed for vision correction. They can be subdivided in several groups according to their consistency (rigid, semi-rigid, soft) and the polymers used (e.g., poly methyl methacrylic acid, copolymer of hydroxyl ethyl methacrylic acid: poly (vinyl pyrrolidone)). The drugs can be added to contact lens, allowing an innovative and relevant approach for the treatment of ocular pathologies. Generally, the drug molecules are bound to contact lenses by presoaking them in drug solutions [80]. The drug-loaded contact lenses offer advantages of increasing the drug residence time on the ocular surface and providing sustained drug release. Due to the close proximity of contact lens with the cornea, the drug molecules are available for absorption. Finally, contact lens soaked with drug could offer the highest bioavailability compared to other noninvasive ophthalmic medications, such as eye drop solutions [27]. Addo et al. reported that post lens tear film allows drug release from the lens and enhances their precorneal residence time of at least 30 min. The bioavailability increases to about 50% with contact lens [81].

#### 3.2.2. Ophthalmic Insert

Ophthalmic inserts are solid or semi-solid sterile devices whose size and shape are specially designed to be placed into the cul-de-sac or conjunctival sac of the eye to deliver active ingredients. They offer many advantages among which the increased ocular residence and the extension of the drug release into the eye are the most relevant. They also improve patient compliance due to the reduction of the dosing frequency. The inserts can be classified according to their solubility behavior in two main categories: Soluble and insoluble inserts. Insoluble inserts can be a matrix or reservoir form. After the release of the active ingredient at a predetermined rate, the empty insert must be removed from the eye. Bioerodible inserts do not need removal as these devices are made of polymers that undergo gradual hydrolysis of the chemical bonds and dissolution.

While releasing the drug, inserts can be considered as technical advances for the ocular delivery of drugs. To our knowledge and particularly concerning the ocular anterior segment, only one anti-inflammatory product has reached the market. This is DEXTANZA^®^, a preservative-free resorbable hydrogel insert containing 0.4 mg of dexamethasone (Table 4). DEXTANZA^®^ was the first FDA-approved intracanalicular insert, a novel route of administration that delivers drug to the surface of the eye. The product originally received FDA approval in November 2018 for the treatment of ocular pain following ophthalmic surgery. Recently, DEXTANZA^®^ received a FDA supplemental new drug application for the therapeutic management of ocular inflammation [27,80,82].

#### 3.2.3. Micro and Nanocarriers for Ocular Drug Delivery

Despite many efforts made by galenic scientists and pharmaceutical companies, effective commercially available drugs to manage affections of the anterior segment of the eye remain a challenge. In this context, nano- and microparticle-based ocular formulations could offer several improvements, such as a substantial increase in the residence time and bioavailability, which are the main limitations of the conventional ocular dosage forms. Therefore, the literature now reports extensive research on several types of micro- and nanoparticle carriers developed for topical ophthalmic administration in order to enhance drug release and improve the bioavailability through the biological membranes of the anterior segment of the eye [83]. Particularly concerning nanosystems, the use of different biocompatible materials (phospholipids, polymers, dendrimers, cyclodextrins, lipids, proteins) made it possible to propose liposomes, nanoparticles, nanosuspensions, nanowafers, nanosponges, nanoemulsions, and nanomicelles as tools with auspicious outcomes for topical ocular delivery of drugs [84]. Table 8, Table 9, and Table 10 group several micro- and nano-formulations of SAIDs, NSAIDs, and immunosuppressive agents described in the literature. Interesting research and review articles have been published, highlighting the benefits of nanosystems in optimizing ocular administration of active ingredients [27,85,86,87]. In some of these publications, the nanocarriers have been studied from several points of view: Composition, physicochemical characteristics, association and release of active ingredients, and potential interests in ocular use. Many potential benefits are therefore expected from ophthalmic topical nanocarriers.

First of all, they may enhance the solubility of hydrophobic drugs. As an example, Jansook et al. formulated dexamethasone with γCD and HPγCD-poloxamer under the form of nanoaggregates, which further exhibit a 15-fold higher concentration than the marketed formulation [88]. As well, Shen et al. and Yu et al. investigated methoxypoly(ethylene glycol)-poly(lactide) polymer (mPEG-PLA) micelles as alternative vehicles for the solubilization and delivery of CsA to the eye [89,90].

Their second advantage is their ability to improve precorneal retention through adhesive properties and active uptake by the corneal and conjunctival epithelia, leading to enhanced ocular permeation [86] in order to produce a rapid effect. Gonzalez-Pizzaro et al. investigated the benefits of a nanoparticulate formulation of fluorometholone based on PLGA and Pluronic 188 in pigs. The nano-formulation was administered 30 min after the induction of ocular inflammation and was found to produce a greater anti-inflammatory effect up to 120 min compared to ISOPTOFLUCON^®^, an eye drop suspension of fluorometholone 1mg/mL (Alcon, Barcelona, Spain), as measured by the ocular inflammation score according to a Draize-modified scoring system. This could be attributed to greater and faster transcorneal permeation [91]. Furthermore, Baba et al. suggested a 50-fold greater ocular penetration of fluorescein diacetate for nanoparticles of hydrolysable dye compared to microparticles [92]. Moreover, Liu et al. described prolonged ocular retention, enhanced corneal permeation, and improved tear production when CsA was loaded in cationized hyaluronic acid-coated spanlastics elastic vesicles made of nonionic surfactants [93]. Indeed, the presence of positive charges in the nanosystems promotes the residence time and the ocular bioavailability of CsA as already described [6] in the case of cationic nanoemulsion (NOVASORB^®^ formulation), licensed in France as IKERVIS^®^ (Santen, Tampere, Finland) and in Europe as VERKAZIA^®^ (Santen, Tampere, Finland). These phenomena are due to the electrostatic interactions between the positively charged droplets and negatively charged mucus protein of the corneal epithelium [74]. This mechanism of action would work in conjunction with the hypothesized reservoir effect of the tear film lipid layer. The combination of these effects, as well as the higher dosage strength, could very likely explain the difference in the dosing regimen between once-a-day IKERVIS^®^ versus twice-a-day RESTASIS^®^ [6].

The third attribute of nanoparticles is their capacity to enhance drug bioavailability by increasing their residence time at the desired sites [94] in order to prolong the effect. Therefore, *N*-trimethyl chitosan nanoparticles encapsulating diclofenac sodium showed a 2.5-fold increase in AUC and a sustained residence time, and the therapeutic concentration was detected up to 12 h in the aqueous humor of rabbit as compared with the marketed formulation [95].

Moreover, coating nanoparticles with positively charged bioadhesive polymers is a strategy designed to enhance the interaction between nanoparticles and the negative charges on the corneal surface and to increase the precorneal residence time and absorption of drug. Chitosan is the most widely used cationic polymer because of its unique properties, such as acceptable biocompatibility, biodegradability, and the ability to enhance the paracellular transport of drugs [96]. Badawi et al. demonstrated in vivo that indomethacin chitosan-coated nanoparticles were able to contact intimately with the cornea, providing slow and gradual indomethacin release with long-term drug levels, thereby increasing delivery to both the external and internal ocular tissues [97].

Note that the use of tacrolimus and ciclosporin-loaded micro or nanoparticles mainly concerns their immunosuppressive activity to prevent immunologic graft rejection [98,99].

In Table 8 and Table 9, data from selected studies are described quickly in terms of the biocompatibility, entrapment efficiency, transcorneal permeation of drug, aqueous humor’s drug concentration, and anti-inflammatory effect in vitro and/or in vivo in the scope of the management of anterior segment inflammation.

Despite the fact that nanocarriers for ocular drug delivery appear very promising for the treatment of the anterior segment of the eye, only two clinical trials are reported on ClinicalTrials.gov. In addition to the technological effort that needs to be overcome for the large-scale manufacture of these nanosystems, the complexity of the dossiers to be submitted to the authorities for placing the products on the market is a limiting factor, in particular concerning the toxicological aspects.

#### 3.2.4. Combined Strategies

The last decades were extremely fructiferous regarding therapeutic developments for ocular disease treatments. Particularly, the incorporation of drug’s nanocarrier into a polymer matrix creates a system that combines the advantages of a micro or nanocarrier and gel:-Convenient administration with good tolerance;-Protection of the drug from the enzymatic metabolism present in the tear film [91,147,171];-Longer retention time on the ocular surface [172];-Sustained release [173];-Bioavailability improvement [174]; and-An increase in the drug’s penetration in the anterior and posterior segments of the eye.

The most used polymers are alginates, chitosan, cellulose derivatives, poloxamer, hyaluronic acid, and carbomer. They are mainly used in order to prolong the retention time on the ocular surface as observed with the gamma scintigraphy study of Gupta et al. The authors demonstrated that PLGA nanoparticles of levofloxacin incorporated in chitosan have good spreading, better retention on the eye, and finally present better bioavailability than the marketed formulation [175].

As seen previously, positively charged bioadhesive polymers can be used to enhance interaction with the negative charges on the corneal surface and to increase the precorneal residence time and drug absorption. Overall, these systems constitute a suitable strategy for the delivery of drugs in order to enhance the drug’s bioavailability. As an example, Ibrahim et al. demonstrated that their nanoparticles included in gels, made of chitosan or poly-ε-caprolactone, showed a 4.8- to 29.7-fold increase in the celecoxib bioavailability compared to celecoxib suspension in rats. The improved bioavailability was indicated in extent and in duration compared with the marketed formulation. On the one hand, this may be due to the high viscosity and bioadhesive properties of chitosan, which prevent the rapid drainage of the formulations and so increased their contact time with the ocular surface. On the other hand, celecoxib-loaded nanoparticles act as drug reservoirs for sustained drug release. Furthermore, these combined formulations increased the celecoxib concentration in both the anterior and posterior segments of the eye. This penetration-enhancing property might be due to chitosan and this ability to open the tight junctions and to increase the permeability of the cell membrane [174].

As previously described, we identified published reports on the micro or nano-delivery systems of NSAIDs and SAIDs combined with polymers for topical ophthalmic administration through a systemic search of PubMed from inception until September 2019. We examined the retrieved reports and included in this review those that presented preclinical research on micro or nanocarriers combined with polymer. In Table 10 and Table 11, selected formulations are described quickly in terms of the biocompatibility, entrapment efficiency, transcorneal permeation of the drug, aqueous humor’s drug concentration, and anti-inflammatory effects.

## 4. Current Biopharmaceutical Attributes of Topical Ophthalmic Formulations 

Regulatory specifications of topical ophthalmic preparation are very restrictive concerning the tolerance, stability, and sterility, with regard to the fragility of the eye. Ophthalmic formulations are also complex, adapted to the specific requirements, and must be well characterized. The examinations that must be performed to determine the properties of each formulation can be divided into two main categories: Sterility testing and physicochemical characterization and biological evaluations. 

### 4.1. Sterility Tests and Physicochemical Attributes

#### 4.1.1. Sterility Tests

Sterility is one of the essential requirements for drug dosage forms applied on the eyeball. The sterility assay is well described in the 2.6.1 monograph of European Pharmacopoeia and USP <71> sterility tests. It involves inoculation in aseptic conditions of the sample examined on two microbiological media:-Fluid thioglycolate medium with resazurin, used for the growth of aerobic and anaerobic bacteria incubated at 30–35 °C; and-Soy-bean casein digest medium, used for the growth of aerobic bacteria and fungi incubated at 20–25 °C.

The samples are incubated for a time not shorter than 14 days. Two methods are described: Direct inoculation or membrane filtration. The number of containers to be tested is fixed by the Pharmacopoeia: 5% of the batch, and a minimum of 2 and maximum of 10 containers per media. The minimum quantity of each container to be tested is also fixed, as an example for liquids: Half of the contents of each container but not less than 1 mL per media.

Finally, the sterility assay is compliant if no growth of microorganisms occurs at 14 days. The procedure for the sterility assay must be performed previously by a suitability test method. The aim of this test is to prove that the drug does not exhibit microorganism growth. As described below, the product must be examined using exactly the same methods. After transferring the content to be tested, an inoculum of a small number of viable microorganisms is added to the media. The inoculums must be < 100 CFU of *Pseudomonas aeruginosa*, *Clostridium sporogenes*, *Staphylococcus aerus*, *Baccilus subtilis*, *Candida albicans*, and *Aspergillus brasiliensis* per media. The incubation for this test is no more than 5 days.

#### 4.1.2. Clarity Examinations

Clarity examination involves the visual assessment of the formulation in suitable lighting on a white and black background. It is well described in Pharmacopoeia and is performed for liquid forms, with the exception of suspensions. This examination applies to eye drops and in situ gels before and after gelling [192].

#### 4.1.3. Osmolality and PH

Osmolality can be measured by the freezing-point depression method. The pH is most often determined using a potentiometric method. pH and osmolality acceptance are 3–8 and 250–450 mOsm/kg for topical ophthalmic administration [193].

#### 4.1.4. Rheological Characterization

The rheological characteristics of ophthalmic formulations are examined at high shear rates using continuous shear techniques and in the viscoelastic region using oscillation techniques. These experiments are currently performed with controlled stress using a cone and plate geometry and the temperature is controlled by a Peltier plate.

The steady-state flow experiments are performed in the range of 0.11 to 100 s^−1^. The frequency sweep method is usually performed between 0.1 and 10 Hz, with shear strain, while the table of shear rate method is performed by increasing the shear rate from 0.1 to 100 s^−1^ or more. The shear stress is measured by this method and the apparent viscosity is calculated by dividing the shear stress by the shear rate. If the relationship between the shear stress and the strain rate is linear, the fluid is Newtonian. If it is non-linear, the fluid is non-Newtonian.

Oscillation frequency tests can be realized over a frequency range of 0.1–10 Hz at a constant stress amplitude under the linear viscoelastic region (5 Pa), which was previously determined by the oscillation stress sweep tests. From the results of the oscillation frequency, the G’ and G” modulus are obtained. If the G’ modulus is greater than G”, the gel exhibits viscous-like mechanism spectra; a contrario, if the G” modulus is greater than G’, the gel exhibits fluid-like mechanism spectra.

The rheological parameter can influence the bioavailability of drugs and the comfort after instillation. The fluids or solutes are eliminated from tears in a few minutes, which results in a short contact time with the eye and high drainage rates and bioavailability for the drugs. To increase the residence time, the viscosity can be increased from 10 to 100 mPa.s, but it may cause discomfort due to blurred vision, foreign body sensation, and damage to the ocular epithelia due to an increase in the shear stress during blinking, resulting in faster elimination due to reflex tears and blinks [194].

#### 4.1.5. Mucoadhesion Tests

There are many methods that have been developed for mucoadhesion measurements. Some are similar to the in vivo situation and are useful when comparing different materials and formulations to find out which may give the longest residence time. Others have been employed to study the mechanisms of mucoadhesion. The usefulness of the different methods depends on the characteristics of the dosage form and what kind of information is being sought.

Some in vivo methods assess the residence time at the application site using gamma scintigraphy, positron emission tomography, or fluorescence, while others involve measurement of the transit time using radioisotopes or fluorescence. The successful use of tracers added to the formulation relies upon the properties of the vehicle remaining unchanged and, therefore, behaving in a manner that is identical to that in the absence of the tracer. So, the results obtained are a genuine reflection of the residence time of the dosage form. The low use of in vivo methods may be explained by the fact that they do not distinguish between mucosal adhesion and other factors affecting the residence time; they are also expensive and they are often accompanied by large standard deviations [79].

The mucoadhesion can be evaluated in vitro by viscosity, rheology, and zeta potential measurements [195]. When using these in vitro methods, not only the method must be chosen but also the mucus substrate. It could be either an excised tissue or a mucus preparation.

Mucoadhesiveness can be determined ex vivo using corneal buttons cut out from freshly isolated porcine eye and fluorescence. A fluorophore is added to formulations, dropped on the corneal surface, and then exposed to a continuous stream of normal saline solution at a rate of 10 mL/min for 5 min to 4 h. This continuous irrigation is followed in order to mimic the blink-induced shear stress on the ocular surface. At the end of the pre-determined exposure time, cryostat sections of the cornea are prepared by embedding the corneal button in optimum cutting temperature compound and are frozen at −20 °C for at least 24 h. The corneal buttons are then sectioned at 5 µm using a cryostat, placed on slides, and imaged for visualization by fluorescence microscopy [196].

#### 4.1.6. Characterization of the Particle Size and Morphology

Multiple methods are used for particle size measurements depending on the size range of particles: Optical microscopy (microscopic particle count test), light obscuration particle count test, dynamic imaging analysis, laser diffraction particle analyzers, electron microscopy (scanning electron microscopy, transmission electron microscopy and atomic force microscopy), DLS (dynamic light scattering), Coulter Counter test, and nanoparticle tracking analysis [192].

Suspensions (micro) or colloidal suspensions require a homogenous and monodispersed population of particles in a specific size range, in order to ensure their suitability for in vitro and in vivo applications and their physical stability. With respect to particle size distribution characterization, a parameter used to define the size distribution is called the “polydispersity index” (PDI). According to the European Pharmacopeia 10th (EP) or the US Pharmacopeia (USP), the ophthalmic preparation meets the requirements if the average number of particles present in the units tested does not exceed the appropriate value listed in the Table 12.

The morphology of particles can be examined by transmission electron microscopy (TEM) or scanning electron microscopy with negative staining. Briefly, the samples are prepared by wetting a carbon-coated copper grid with a small drop of diluted formulation (5–10 μL). Upon drying, they are stained with 1% uranyl acetate and 2% phosphotungstic acid, air-dried at room temperature, and viewed by TEM. Imaging viewer software is used to perform the image capture and analysis [197]. CryoTEM is also used [198].

#### 4.1.7. Zeta Potential Measurement

The electrophorectic mobility of nanoparticles is determined by using a Zetasizer and transformed into the zeta potential by using the Smoluchowski equation [199].

#### 4.1.8. Drug and Preservative Contents

The drug and preservative contents must be determined by an analytical drug quantification methodology and validated according to International Council for Harmonisation of Technical Requirements for Pharmaceuticals for Human Use (ICH) Q2 (R1) guidelines in order to evaluate the specificity, linearity, repeatability, intermediate fidelity, limit of detection (LOD), and limit of quantification (LOQ) [200]. The most frequently used method is HPLC [192].

If it is a nanoparticulate formulation, the entrapment efficiency (EE%) must be determined. The EE% is found by subtracting the free drug from the total concentration found in the nanosuspension [192].

#### 4.1.9. Stability Study

ICH Q1A (R2) defines the stability data package for a new drug substance or drug product that is sufficient for a registration application within the three regions of the EC, Japan, and the United States.

The purpose of stability testing is to provide evidence on how the quality of a drug substance or drug product varies with time under the influence of a variety of environmental factors, such as the temperature, humidity, and light, and to establish the test period for the drug substance or a shelf life for the drug product and recommended storage conditions. The stability studies should include testing of those attributes of the drug product that are susceptible to change during storage and are likely to influence the quality, safety, and/or efficacy.

The testing should cover, as appropriate, the physical, chemical, biological, and microbiological attributes; and the preservative content (e.g., antioxidant, antimicrobial preservative). A stability study consists of following these parameters at different pre-determined times (e.g., T0 and 3, 6, 9, and 12 months) and stored in one or more controlled temperatures and humidity. An approach for analyzing the data on a quantitative attribute that is expected to change with time is to determine the time at which the 95% one-sided confidence limit for the mean curve intersects the acceptance criterion [201].

#### 4.1.10. In Vitro Drug Release Study

In vitro release characteristics can be investigated using the dialysis membrane, whose molecular weight cut-off is between 1000 and 14,000, in the Franz cell [124] or modified rotating paddle apparatus [192]. The release medium is generally made of phosphate-buffered solution (PBS, pH 7.4), and sometimes, PBS contains polysorbate 80 in order to facilitate the drug’s solubilization by increasing its wettability in PBS and to maintain sink conditions. The dialysis membrane and cell are maintained at 35–37 °C. At predetermined times, a volume of release medium is withdrawn and samples are measured [202].

Finally, other specific tests may be useful, according to the pharmaceutical form. For example, the gelification ability to form the gel in contact with the eye must be assess for an in situ gelling system or the swelling index for inserts [192].

### 4.2. Biological Evaluations

#### 4.2.1. Toxicity and Biocompatibility Tests

Corneal damage results from irritation and inflammation, causing mild discomfort to tissue corrosion, and resulting in irreversible blindness. During drug evaluation, the eye irritation potential and eye toxicity of eye drops must be tested to ensure the safety of the drug product before clinical trials in humans.

##### In Vitro Tests

In vitro testing models using cultured cells area present numerous advantages compared to in vivo or ex vivo testing as they are relatively inexpensive, simple, and quick to implement.

Most in vitro ocular toxicity assays consist of a monolayer of cultured cells and a cytotoxicity assessment in response to a test material. Among the methods of assessing cytotoxicity are the MTT assay, LDH assay, fluorescein leakage tryptan blue exclusion, fluorescent staining with propidium iodide, and neutral red uptake/release tests or ALAMAR^®^ BLUE assay [203]. Each of these methods has their advantages and limitations. In general, a combination of two or more of these methods is normally used to assess cytotoxicity.

For example, the MTT assay in a short time exposure (STE) according to the Organization for Economic Cooperation and Development (OECD) guideline is performed after a 24-h stabilization of the cells, then fresh medium containing either different concentrations (5% and 0.05%) of the formulation, blank, or formulation without drug are added. Cells are incubated for 5 min at 37 °C in order to compare the cytotoxicity of different concentrations and incubation times on cells. After incubation, media is removed and fresh medium and MTT solution are added to each well. Incubation is allowed for another 4 h in darkness at 37 °C. Since living cells metabolize the MTT and form blue formazan crystals, DMSO is added to dissolve the formazan crystals. Absorbance may be read with any filter in the wavelength range of 550–600 nm, and the percentage of viability can be calculated. The viability of the treated cell cultures is expressed as a percentage of the control untreated cell cultures assumed to be 100%. According to OECD, Table 13 summarizes the prediction model of STE [204]. Note that the United Nations Globally Harmonized System of Classification and Labelling of Chemicals (UN GHS) is a system proposing the classification of chemicals (substances and mixtures) according to standardized types and levels of physical, health, and environmental hazards. This system addresses corresponding communication elements, such as pictograms, signal words, hazard statements, precautionary statements, and safety data sheets. UN GHS Category 1corresponds to “Serious eye damage”, UN GHS Category 2 corresponds to “Eye irritation”, and finally, UN GHS No Category corresponds to chemicals that are not classified as UN GHS Category 1 or 2 (2A or 2B).

It is an ethical alternative to in vivo studies but do not represent the variability observed in animal and human trials. Generally, in vitro cell culture models can be classified into three different groups, namely primary cell cultures, immortalized cell lines, and reconstructed tissue cultures. According to Rökkö et al., Table 14 summarizes the advantages and disadvantages of each type of cell. The most frequently used cells are Y79 [180], HEK 293 [177], SIRC [204], or HCEC [205].

In vitro assays and models provide useful data that complement in vivo studies, allowing for significant reductions in the numbers of animals used.

Numerous in vitro methods are used to predict the biocompatibility or irritation effects of formulations for topical administration, according or not to OECD guidelines: Reconstructed human cornea-like epithelium eye irritation test, fluorescein leakage test method, VITRIGEL^®^ EIT method, EPIOCULAR^®^ time to toxicity, OCUL^®^ IRRITECTION, and the neutral red release or red blood cell test [206].

##### Ex Vivo Tests

Several ex vivo models have been developed as excised rabbit, porcine, or bovine corneas, since human corneas are generally reserved for transplant purpose only. They exhibit interspecies variations due to differences in their anatomy and morphology; however, with some caution, they can be used to establish good qualitative comparisons of different drug transport pathways.

Rabbit eyes, although smaller than human eyes, are the most preferred for ex vivo models as they can also conveniently be used for in vivo studies, facilitating ex vivo–in vivo correlations. As rabbit eyes lack Bowman’s layer, thus penetration is generally much higher and cannot be correlated well to humans.

Pig eyes are structurally the most similar to human eyes in terms of the globe size, corneal thickness, globe diameter to corneal length ratio, and the presence of the Bowman’s layer.

Bovine eyes, on the other hand, are significantly larger than human eyes, and the corneal epithelium is almost twice as thick. Furthermore, it is important to keep in mind that human and animal corneas may significantly differ in the metabolic enzymes and transporters present on their surface, thus affecting the bioavailability [207].

To date, neither an in vitro nor an ex vivo test is capable of classifying chemicals as the Draize test. Currently, only a limited number of ocular toxicity assays have resulted in validation and regulatory acceptance: Bovine corneal opacity and permeability (BCOP), isolated chicken eye (ICE), fluorescein leakage (FL), and short time exposure (STE) tests have been accepted by ICCVAM and OECD.

##### In Vivo Tests

Live animals have been used to assess and evaluate potentially harmful products to eyes since the 18th century. The international standard assay for acute toxicity is the rabbit in vivo Draize eye test, which was developed in the 1940s by the Food and Drug Administration (FDA). New Zealand white (NZW) rabbits are the most commonly used. The procedure involves the application of 0.1 mL (or 0.1 g solid) of the test substance onto the cornea and cul-de-sac conjunctival of one eye of a conscious rabbit for up to 72 h while the other eye serves as the untreated control [208]. The original Draize protocol used at least six rabbits per test, but this was reduced to three animals or a single when serious ocular damage is expected, those with severe lesions being “humanely” euthanized. The latest Draize test guidelines, including the application and delivery of analgesics and anesthetics, was introduced in 2012 [209] to reduce animal pain and suffering. The rabbits are observed at selected intervals for up to 21 days for signs of irritation, including redness, swelling, cloudiness, edema, hemorrhage, discharge, and blindness [203]. In fact, the Draize testing is the only test formally accepted and validated to assess the full range of irritation severity. Both reversible and irreversible ocular effects can be identified using this test [210].

The observed degree of irritancy allows classification of the substances, based on the subjective scoring of the effect on the cornea, conjunctiva, and iris, ranging from non-irritating to severely irritating.

Despite its “gold standard” status, it is often criticized due to its subjective and time-consuming nature, lack of repeatability, variable estimates, insufficient relevance of test chemical application, high dosages, and over-prediction of human responses primarily due to interspecies differences [211,212]. For many years, the legislation of many countries is the European directive 2010/63/EU, which tries to reduce, refine, and replace animal testing in biological experiments and promote alternatives. However, the reduction of animal use is primarily concentrated on toxicology studies since no government agency to date has eliminated animal use in basic pharmaceutical development.

One of the alternative in vivo tests is the low-volume eye-irritation test (LVET). It was developed in response to a recommendation from the national research council [213]. It is a refinement of the Draize test with a lower volume: 0.01 mL/0.01 g applied on the corneal surface of the right eye of the animal without forced eyelid closure employed and not on the conjunctival sac. It is less stressful for the animal. However, the LVET is still criticized for the use of an animal and the risk of false negative results and it is not considered to be a valid replacement nor recommended for prospective ocular safety testing [211].

##### In Silico Tests

In silico models are computer-generated models that can play a useful role in predicting the ocular toxicity of a substance, using quantitative structure–activity relationships (QSARs) [211].

#### 4.2.2. Pharmacokinetic Studies

For ocular drug products, there is no requirement for pharmacokinetics studies in human subjects. This is because the relevant target or surrogate tissues cannot be sampled serially. For the same reasons, during development, pharmacokinetics data rely on the use of animal’s models, such as the rabbit, monkey, dog, and pig.

##### Ex Vivo Transcorneal Permeation Studies

Transcorneal permeation studies are carried out by putting the eye drops (0.4 to 1 mL) on a freshly excised cornea. The cornea is freshly excised and fixed between the clamped donor and receptor compartments of an all-glass modified Franz diffusion cell in such a way that its epithelial surface faces the donor compartment. The receptor compartment is filled with freshly prepared simulated tear fluid (pH 7.4). The permeation study is carried out for 4 h, and samples are withdrawn from the receptor and analyzed. At the end of the experiment, the corneal hydration of each cornea must be evaluated. Different excised cornea can be used, such as bovine, porcine, rabbit, goat, sheep, or buffalo [214].

##### In Vivo Tests

The most used in vivo pharmacokinetics tests are tear fluid or aqueous humor sampling [141]. Some protocol evaluate the pharmacokinetics of the drug in eye tissues but animals need to be euthanized [174]. The pharmacokinetics study is also conducted using a single-dose-response design. Rats are used to evaluate uveitis while rabbits are used to evaluate conjunctivitis. The animals are divided in two groups: Verum and control. The animals are lightly sedated. Each formulation is instilled into the inferior conjunctival sac of the right eyes of the animals, whereas the left eyes serve as the control by application of the plain dosage form. The eyes are held open for at least 20 s to allow for adequate ocular surface contact of the formulations and to prevent excessive blinking during application of the dosage form, and then the eyelids are held together for an additional 10 s to avoid rapid loss of the formulations. Part of the animals are euthanized at a predetermined time and then scarified by thoracic opening. Blood samples are collected.

Both eyes are enucleated and dissected while fresh to separate different eye tissues of the cornea, conjunctiva, anterior sclera, aqueous humor, lens, iris, vitreous body, and posterior eye cup. The amount of drug retained from the different parts of the eye must be further quantified [174].

Some in vivo methods assess transcorneal permeation by radiolabelling and imaging by gamma scintigraphy [124] or positron emission tomography [215]. The successful use of tracers added to the formulation relies upon the properties of the vehicle remaining unchanged and, therefore, behaving in a manner that is identical to that in the absence of the tracer so that the results obtained are a genuine reflection of the residence time of the dosage form. The low use of in vivo methods may be explained by the cost and the large standard deviations of the method [79].

Another alternative approach includes microdialysis. The microdialysis probe is generally placed in the liquid compartments of the eye, such as the aqueous humor and vitreous humor, and thus allows continuous sampling, making it possible to access pharmacokinetic parameters.

#### 4.2.3. Efficacy Testing

The anti-inflammatory efficacy test for topical ophthalmic formulations consists in administering a proinflammatory substance to animals, i.e., carrageenan [188] or arachidonic acid, and a more specific induced inflammation model exists, such as autoimmune uveitis [216] or ethanol burn [162].

Usually, rabbits are used for conjunctivitis and rats for uveitis [100,105]. Inflammation is induced to a marked extent one hour after carrageenan injection and 30 min after sodium arachidonate instillation.

For example, a usual protocol consists in comparing the formulation to a commercial drug and control group (NaCl 0.9% or BSS). The assay is carried out using New Zealand albino male rabbits (*n* = 6 /group). The study is conducted with the application of 50 µL of 0.5% sodium arachidonate dissolved in PBS in the right eye, using the left eye as a control. After 30 minutes of exposure, 50 µL of each formulation are instilled. In order to evaluate the prevention of inflammation, the evaluation of inflammation is performed from the application of formulation up to 150 min according to the Draize-modified scoring system. It includes histopathological examination, such as inhibition of polymorphonuclear leukocytes’ migration and lid closure scores and the alterations of interleukin IL-17 and IL-10 at mRNA and protein levels in either aqueous humor or serum [168].

## 5. Conclusions

Still today, the ocular administration of drugs remains a huge challenge for ophthalmologists and galenic scientists. This review, mainly devoted to the management of inflammation of the anterior segment of the eye, offers a complete view on the conventional anti-inflammatory products marketed in France, Europe, and the USA. Furthermore, the review highlights the progress of therapeutic efficacy expected with the implementation of new delivery systems. In addition, the main in vitro, ex vivo, and in vivo study methods for the development of ophthalmic anti-inflammatory products were considered. Finally, through the literature cited in this review, scientists have an up-to-date background information to improve the efficacy and tolerability of future topical anti-inflammatory products for the anterior segment of the eye.

## Figures and Tables

**Figure 1 pharmaceutics-12-00570-f001:**
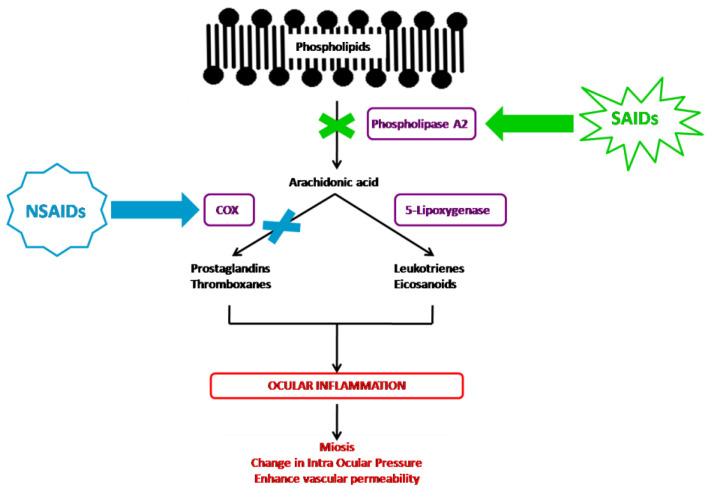
NSAIDs’ and SAIDs’ mechanisms of action in the inflammatory cascade.

**Figure 2 pharmaceutics-12-00570-f002:**
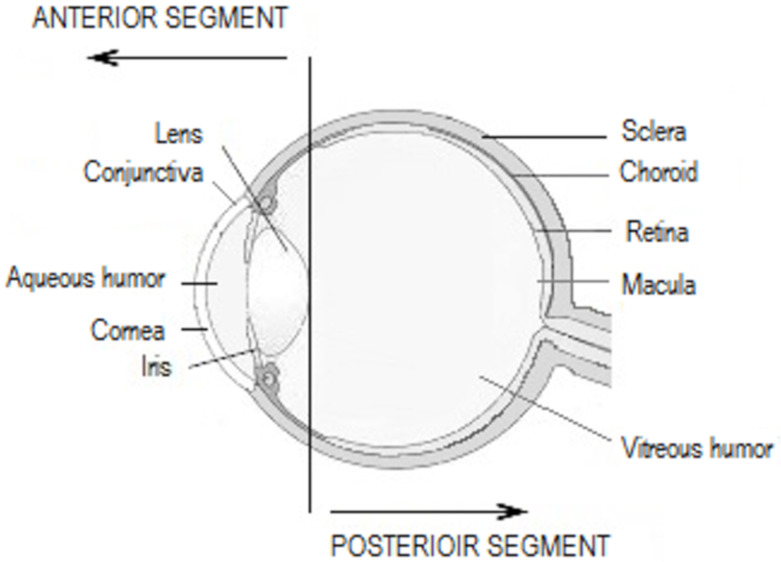
Anatomy of the eye.

**Figure 3 pharmaceutics-12-00570-f003:**
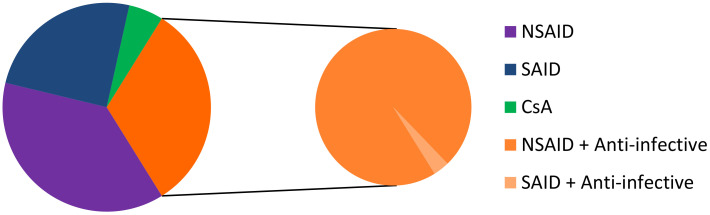
Repartition of NSAID, SAID, CsA, NSAID + anti-infective drugs, and SAID + anti-infective drugs on the USA, European, and French markets. CsA: ciclosporin A.

**Figure 4 pharmaceutics-12-00570-f004:**
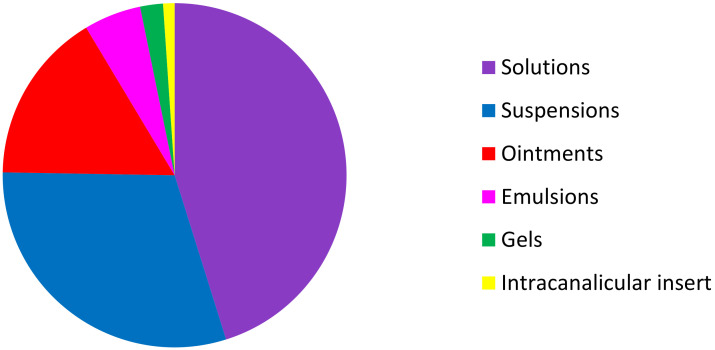
Distribution of the different formulations of NSAID, SAID, CsA NSAID + anti-infective drugs, and SAID + anti-infective drugs on the USA, European, and French markets.

**Table 1 pharmaceutics-12-00570-t001:** Non-steroidal (NSAIDs) and steroidal anti-inflammatory drugs (SAIDS) marketed for oral, parenteral, and topical ophthalmic administrations as of 26th February, 2019, in France, the EU, and the USA, and their chemical classes.

DCI	NSAID/ SAID	Chemical Classes	Routes of Administration
Aceclofenac	NSAID	Aryl-acetic acid derivatives	Per os
Alminoprofen	NSAID	Propionic acid derivatives	Per os
Betamethasone	SAID		Per os, inj and topical ophthalmic
Bromfenac	NSAID	Aryl-acetic acid derivatives	Topical ophthalmic
Celecoxib	NSAID	Selective cylooxygenase -2 inhibitors	Per os
Deflazacort	SAID		Per os
Dexamethasone (base and phosphate sodium)	SAID		Per os, inj and topical ophthalmic
Dexketoprofen	NSAID	Propionic acid derivatives	Inj
Diclofenac	NSAID	Aryl-acetic acid derivatives	Per os and topical ophthalmic
Difluprednate	SAID		Topical ophthalmic
Etodolac	NSAID	Indole and indene derivatives	Per os
Etoricoxib	NSAID	Selective cylooxygenase -2 inhibitors	Per os
Fluorometholone (base and acetate)	SAID		Topical ophthalmic
Flurbiprofen	NSAID	Propionic acid derivatives	Topical ophthalmic
Hydrocortisone	SAID		Per os, inj and topical ophthalmic
Ibuprofen	NSAID	Propionic acid derivatives	Per os and inj
Indomethacin	NSAID	Indole and indene derivatives	Per os, inj and topical ophthalmic
Ketoprofen	NSAID	Propionic acid derivatives	Per os and inj
Ketorolac tromethamine	NSAID	Aryl-acetic acid derivatives	Per os, inj and topical ophthalmic
Loteprednoletobonate	SAID		Topical ophthalmic
Meclofenamate sodium	NSAID	Fenamic acid derivatives	Per os
Mefenamicacid	NSAID	Fenamic acid derivatives	Per os
Meloxicam	NSAID	Enolic acid derivatives	Per os and inj
Methylprednisolone	SAID		Per os and inj
Nabumetone	NSAID	Non acidic derivatives	Per os
Naproxen	NSAID	Propionic acid derivatives	Per os
Nepafenac	NSAID	Aryl-acetic acid derivatives	Topical ophthalmic
Niflumic acid	NSAID	Fenamic acid derivatives	Per os
Oxaprozin	NSAID	Propionic acid derivatives	Per os
Parecoxib	NSAID	Selective cylooxygenase -2 inhibitors	Inj
Piroxicam	NSAID	Enolic acid derivatives	Per os and inj
Prednisolone (Acetate and Sodium Phosphate)	SAID		Per os, inj and topical ophthalmic
Prednisone	SAID		Per os
Salicylic Acid	NSAID	Salicylic acid derivatives	Per os and topical ophthalmic
Sulindac	NSAID	Indole and indene derivatives	Per os
Tenoxicam	NSAID	Enolic acid derivatives	Per os
Tiaprofen	NSAID	Propionic acid derivatives	Per os
Tolmetin	NSAID	Aryl-acetic acid derivatives	Per os
Triamcinolone	SAID		Inj and topical ophthalmic

Per os: NSAIDs or SAIDs actually available for oral administration; inj: NSAIDs or SAIDs actually available for parenteral administration; topical ophthalmic: NSAIDs or SAIDs actually available for topical ophthalmic administration.

**Table 2 pharmaceutics-12-00570-t002:** The USA-, European- and French-marketed NSAID, SAID, and CsA medicines listed as of 26th February, 2019 for topical use in ophthalmology.

DCI	NSAID/SAID	Product Names in USA, EU and France
**Bromfenac**	NSAID	BROMSITE EQ^®^, PROLENSA EQ^®^, YELLOX^®^
**Ciclosporin A**	CsA	CEQUA^®^, IKERVIS^®^, RESTASIS®, RESTASIS^®^ MULTIDOSE, VERKAZIA^®^
**Dexamethasone** **(Base or Sodium Phosphate)**	SAID	CHIBRO CADRON^®^, DEXAFREE^®^, DEXASPORIN^®^, DEXTENZA^®^, FRAKIDEX^®^, MAXIDEX^®^, MAXIDROL^®^, MAXITROL^®^, STERDEX^®^, TOBRADEX^®^
**Diclofenac**	NSAID	VOLTAREN^®^, VOLTAREN^®^OPHTA, VOLTAREN^®^OPHTABAK
**Difluprednate**	SAID	DUREZOL^®^
**Fluorometholone** **(Acetate or Base)**	SAID	FLUCON^®^, FML^®^, FML FORTE^®^ FLAREX^®^
**Flurbiprofen**	NSAID	OCUFEN^®^
**Hydrocortisone**		SOFTACORT^®^
**Indomethacin**	NSAID	INDOCOLLYRE^®^, INDOBIOTIC^®^
**Ketorolac tromethamine**	NSAID	ACULAR^®^, ACULAR LS^®^, ACUVAIL^®^,
**Loteprednol etabonate**	SAID	ALREX^®^, INVELTYS^®^, LOTEMAX^®^, LOTEMAX SM^®^, ZYLET^®^
**Nepafenac**	NSAID	ILEVRO^®^, NEVANAC^®^
**Prednisolone** **(Acetate or Sodium Phosphate)**	SAID	BLEPHAMIDE^®^, BLEPHAMIDE S.O.P^®^,OMNIPRED^®^, PRED FORTE^®^, PRED MILD
**Salicylicacid**	NSAID	ANTALYRE^®^, CIELLA^®^
**Triamcinolone**	SAID	CIDERMEX^®^

**Table 3 pharmaceutics-12-00570-t003:** Sites of action in the anterior segment of the eye and therapeutic use of the widely prescribed anti-inflammatory and immunosuppressive drugs.

Indications			Commonly Used Drugs
**Management of post-operative inflammation**			Diclofenac, ketorolac, SAIDs [1,53]
**Prevention of intra-operative miosis**			Flurbiprofen, ketorolac [54]
**Anterior segment**	**Conjunctiva**	Treatment of allergic conjunctivitis	Ketorolac, SAIDs [1,53]
		Treatment of blepharoconjunctivitis	SAIDs [53]
		Treatment of corneo-conjunctival burn	SAIDs [53]
		Increase tear production in patients with keratoconjunctivitis sicca associated with dry eye syndrome	Ciclosporin [55]
	**Cornea**	Prevention of corneal graft rejection	Dexamethasone [52,53]
		Control of pain after refractive surgery	Diclofenac, ketorolac [1]
		Treatment of immune keratitis	SAIDs [53]
		Treatment of bacterial keratitis	SAIDs [53]

**Table 4 pharmaceutics-12-00570-t004:** Topical ocular pharmaceutical forms and compositions of SAID-containing medicines in the US, European, or French markets listed as of 26th February, 2019.

Trade Name and Presentation	Active Substance	Excipients	Pharmaceutical Form	Marketed In	Year of Authorization
**Alrex 0.2%**	Loteprednol etabonate	Benzalkonium chloride,	suspension/drops	USA	1998
Multidose bottle 2.5, 5 and 10 mL		Edetate disodium, glycerin, povidone, purified water, tyloxapol, hydrochloric acid and/or sodium hydroxide to adjust the pH			
**Dexafree 0.1%**	Dexamethasone phosphate		solution/drops	Fr	2006
Single use vial 0.4 mL		Edetate disodium, sodium phosphate dibasic, sodium chloride, water for injection			
**Dexamethasone Sodium Phosphate EQ 0.1% Phosphate**	Dexamethasone phosphate	Sodium bisulfite, phenylethyl alcohol, benzalkonium chloride,	solution/drops	USA	1996
Multidose bottle 5 mL		Sodium citrate, sodium borate, creatinine, polysorbate 80, edetate disodium dihydrate, purified water, hydrochloric acid			
**Dextenza 0.4MG**	Dexamethasone		intracanalicular insert	USA	2018
Single dose		4-arm polyethylene glycol (PEG) *N*-hydroxysuccinimidyl glutarate (20 K), trilysine acetate, *N*-hydroxysuccinimide-fluorescein, sodium phosphate dibasic, sodium phosphate monobasic, water for injection			
**DUREZOL 0.05%**	Difluprednate	Sorbic acid,	emulsion	USA	2008
2.5 mL in 5 mL multidose bottle 5 mL in 5 mL multidose bottle		Boric acid, castor oil, glycerin, polysorbate 80, purified water, sodium acetate, sodium EDTA, sodium hydroxide to adjust the pH			
**FLAREX 0.1%**	Fluorometholone acetate	Benzalkonium chloride,	suspension/drops	USA	1986
5 mL in 8 mL multidose bottle 10 mL in 10 mL multidose bottle 15 mL in 15 mL multidose bottle		Sodium chloride, monobasic sodium phosphate, edetate disodium, hydroxyethyl cellulose, tyloxapol, hydrochloric acid and/or sodium hydroxide to adjust the pH, purified water			
**FLUCON 0.1%**	Fluorometholone	Benzalkonium chloride,	suspension/drops	Fr	1980
Multidose bottle 3 mL		Monobasic sodium phosphate, dibasic sodium phosphate, polysorbate 80, sodium chloride, edetate disodium, polyvinyl alcohol, hydroxypropylmethylcellulose, hydrochloric acid and/or sodium hydroxide to adjust the pH			
**FML 0.1%**	Fluorometholone	Phenylmercuric acetate,	ointment	USA	1985
3.5 g tube		Mineral oil, petrolatum alcohol, lanolin alcohol, white petrolatum			
**FML 0.1%**	Fluorometholone	Benzalkonium chloride,	suspension/drops	USA	1972
5 mL in 10 mL multidose bottle 10 mL in 15 mL multidose bottle 15 mL in 15 mL multidose bottle		Edetate disodium, polysorbate 80, polyvinyl alcohol, purified water, sodium chloride, sodium phosphate dibasic, sodium phosphate monobasic, sodium hydroxide			
**FML FORTE 0.25%**	Fluorometholone	Benzalkonium chloride,	suspension/drops	USA	1986
5 mL in 10 mL multidose bottle 10 mL in 15 mL multidose bottle 15 mL in 15 mL multidose bottle		Edetate disodium, polysorbate 80, polyvinyl alcohol, purified water, sodium chloride, sodium phosphate dibasic, sodium phosphate monobasic, sodium hydroxide			
**Inveltys 1%**	Loteprednol etabonate		suspension/drops	USA	2018
2.8 mL in 5 mL multidose bottle		Glycerin, sodium citrate dihydrate, poloxamer 407, sodium chloride, edetate disodium dihydrate, citric acid			
**Lotemax 0.5%**	Loteprednol etabonate		gel	USA	2012
5 g in 10 mL multidose bottle		Boric acid, edetate disodium, glycerin, polycarbophil, propylene glycol, sodium chloride, tyloxapol, water for injection, sodium hydroxide to adjust to the pH			
**Lotemax 0.5%**	Loteprednol etabonate	Benzalkonium chloride,	suspension/drops	USA	1998
Multidose bottle 2.5, 5, 10 and 15 mL		Edetate disodium, glycerin, povidone, purified water, tyloxapol, hydrochloric acid and/or sodium hydroxide to adjust the pH			
**Lotemax 0.5%**	Loteprednol etabonate		ointment	USA	2011
3.5 g tube		Mineral oil, white petrolatum,			
**Lotemax 0.5%**	Loteprednol etabonate		gel	USA	2012
5 g in 10 mL multidose bottle		Boric acid, edetate disodium, glycerin, polycarbophil, propylene glycol, sodium chloride, tyloxapol, water for injection, sodium hydroxide to adjust to the pH			
**Lotemax Sm 0.38%**	Loteprednol etabonate	Benzalkonium chloride,	gel	USA	2019
5 g in 10 mL multidose bottle		Boric acid, edetate disodium dihydrate, glycerin, hypromellose, poloxamer, polycarbophil, propylene glycol, sodium chloride, water for injection,			
**Loteprednol Etabonate 0.5%**	Loteprednol etabonate	Benzalkonium chloride,	suspension/drops	USA	2019
Multidose bottle 5, 10 and 15 mL		Edetate disodium, glycerin, povidone, purified water, hydrochloric acid and/or sodium hydroxide to adjust the pH, tyloxapol			
**Maxidex 0.1%**	Dexamethasone	Benzalkonium chloride,	suspension/drops	Fr	1992
Multidose bottle 3 mL		Sodium phosphate monobasic, polysorbate 80, edetate disodium, sodium chloride, methylhydroxypropylcellulose, citric acid, purified water			
**Maxidex 0.1%**	Dexamethasone	Benzalkonium chloride,	suspension/drops	USA	1962
Multidose bottle 5 mL		Hypromellose, sodium chloride, dibasic sodium phosphate, polysorbate 80, edetate disodium, citric acid and/or sodium hydroxide to adjust the pH, purified water			
**Omnipred 1%**	Prednisolone acetate	Benzalkonium chloride,	suspension/drops	USA	1973
Multidose bottle 5 and 10 mL		Hypromellose, dibasic sodium phosphate, polysorbate 80, edetate disodium, glycerin, citric acid and/or sodium hydroxide to adjust the pH, purified water			
**Pred Forte 1%**	Prednisolone acetate	Benzalkonium chloride,	suspension/drops	USA	1973
5 mL in 10 mL multidose bottle 10 mL in 15 mL multidose bottle 15 mL in 15 mL multidose bottle		Boric acid, edetate disodium, hypromellose, polysorbate 80, purified water, sodium bisulfite, sodium chloride, sodium citrate			
**PRED MILD 0.12%**	Prednisolone acetate	Benzalkonium chloride,	suspension/drops	USA	1972
5 mL in 10 mL multidose bottle 10 mL in 15 mL multidose bottle		Boric acid, edetate disodium, hypromellose, polysorbate 80, purified water, sodium bisulfite, sodium chloride, sodium citrate			
**Prednisolone Sodium Phosphate EQ 0.9%**	Prednisolone sodium phosphate	Benzalkonium chloride,	solution/drops	USA	1994
5 mL in 10 mL multidose bottle 10 mL in 15 mL multidose bottle 15 mL in 15 mL multidose bottle		Hypromellose, monobasic sodium phosphate, dibasic sodium phosphate, sodium chloride, edetate disodium dihydrate, purified water, hydrochloric acid and/or sodium hydroxide to adjust the pH			
**Softacort 0.335%**	Hydrocortisone		solution/drops	Fr	2017
Single use vial 0.4 mL		Sodium phosphate dibasic, monobasic sodium phosphate, edetate disodium, hydrochloric acid to adjust the pH, water for injection,			

**Table 5 pharmaceutics-12-00570-t005:** Topical ocular pharmaceutical forms and compositions containing NSAID medicines in the US, European, or French markets listed as of 26th February 2019.

Trade Name and Presentation	Active Substance	Excipients	Pharmaceutical Form	Marketed In	Year of Authorization
**Acular 0.5%**	Ketorolac trometamol	Benzalkonium chloride,	solution/drops	Fr	1991
Multidose bottle 5 mL		Sodium chloride, edetate disodium, octoxynol 40, hydrochloric acid and/or sodium hydroxide to adjust the pH, purified water			
**Acular 0.5%**	Ketorolac tromethamine	Benzalkonium chloride,	solution/drops	USA	1992
Multidose bottle 5 and 10 mL		Edetate disodium, octoxynol 40, purified water, sodium chloride, hydrochloric acid and/or sodium hydroxide to adjust the pH			
**Acular LS 0.4%**	Ketorolac tromethamine	Benzalkonium chloride,	solution/drops	USA	2003
Multidose bottle 5 and 10 mL		Edetate disodium, octoxynol 40, purified water, sodium chloride, hydrochloric acid and/or sodium hydroxide to adjust the pH			
**Acuvail 0.45%**	Ketorolac tromethamine		solution/drops	USA	2009
Single use vial 0.4 mL		Carboxymethylcellulose, sodium chloride, sodium citrate, purified water, hydrochloric acid and/or sodium hydroxide to adjust the pH			
**Antalyre 0.1%**	Salicylic acid		solution/drops	Fr	2004
Single use vial 0.4 mL		Borax, boric acid, sodium chloride, purified water			
**Bromfenac Sodium EQ 0.09% Acid**	Bromfenac sodium	Benzalkonium chloride,	solution/drops	USA	2014
1.7 mL in 6 mL multidose bottle		Boric acid, edetate disodium, polysorbate 80, povidone (K30), purified water, sodium borate, sodium sulfite anhydrous, sodium hydroxide to adjust the pH			
**Bromsite EQ 0.075% Acid**	Bromfenac sodium	Benzalkonium chloride,	solution/drops	USA	2016
5 mL in 7.5 mL multidose bottle		Boric acid, sodium borate, citric acid anhydrous, sodium citrate dihydrate, poloxamer 407, polycarbophil, sodium chloride, edetate disodium, sodium hydroxide, water for injection			
**Ciella 0.1%**	Salicylic acid		solution	Fr	2004
Multidose bottle 5 mL		Borax, sodium chloride, boric acid, rose-flavored water, purified water			
**Diclofenac Sodium 0.1%**	Diclofenac sodium		solution/drops	USA	2008
Multidose bottle 2.5 and 5 mL		Polyoxyl 35 castor oil, boric acid, tromethamine, sorbic acid, edetate disodium, purified water			
**Diclofenac Sodium 0.1%**	Diclofenac sodium		solution/drops	USA	2015
Multidose bottle 5 mL		Polyoxyl 35 castor oil, boric acid, tromethamine, sorbic acid, edetate disodium, purified water			
**Diclofenac Sodium 0.1%**	Diclofenac sodium		solution/drops	USA	2007
Multidose bottle 5 mL		Polyoxyl 35 castor oil, boric acid, tromethamine, sorbic acid, edetate disodium, purified water			
**Diclofenac Sodium 0.1%**	Diclofenac sodium		solution/drops	USA	2008
Multidose bottle 5 mL		Polyoxyl 35 castor oil, boric acid, tromethamine, sorbic acid, edetate disodium, purified water			
**Diclofenac Sodium 0.1%**	Diclofenac sodium		solution/drops	USA	2008
Multidose bottle 5 mL		Polyoxyl 35 castor oil, boric acid, tromethamine, sorbic acid, edetate disodium, purified water			
**Flurbiprofen Sodium 0.03%**	Flurbiprofen sodium	Thimerosal,	solution/drops	USA	1995
Multidose bottle 2.5 mL		Citric acid, edetate disodium, polyvinyl alcohol, potassium chloride, purified water, sodium chloride, sodium citrate, hydrochloric acid and/or sodium hydroxide to adjust the pH			
**Ilevro 0.3%**	Nepafenac	Benzalkonium chloride,	suspension/drops	USA	2012
1.7 mL in 4 mL multidose bottle		Boric acid, propylene glycol, carbomer 974P, sodium chloride, guar gum, carboxymethylcellulose sodium, edetate disodium, hydrochloric acid and/or sodium hydroxide to adjust the pH, purified water			
**Indocollyre 0.1%**	Indomethacin	Thimerosal,	solution/drops	Fr	1996
Multidose bottle 5 mL		Arginine, hydroxypropylbetadex, hydrochloric acid, purified water			
**Indocollyre 0.1%**	Indomethacin		solution/drops	Fr	1997
Single use vial 0.35 mL		Arginine, hydroxypropylbetadex, hydrochloric acid, purified water			
**Ketorolac Tromethamine 0.4%**	Ketorolac tromethamine		solution/drops	USA	2009
NA					
**Ketorolac Tromethamine 0.4%**	Ketorolac tromethamine		solution/drops	USA	2009
NA					
**Ketorolac Tromethamine 0.4%**	Ketorolac tromethamine		solution/drops	USA	2009
NA					
**Ketorolac Tromethamine 0.4%**	Ketorolac tromethamine		solution/drops	USA	2018
NA					
**Ketorolac Tromethamine 0.5%**		Benzalkonium chloride,	solution/drops	USA	2009
5 mL in 11 mL multidose bottle 10 mL in 11 mL multidose bottle	Nepafenac Ketorolac tromethamine	Edetate disodium, octoxynol 40, sodium chloride, hydrochloric acid and/or sodium hydroxide to adjust the pH, water for injection	suspension/drops solution/drops	USA USA	2012 2009
**KETOROLAC TROMETHAMINE 0.5%**		Benzalkonium chloride,			
Multidose bottle 5 and 10 mL	Indomethacin Ketorolac tromethamine	Edetate disodium, octoxynol 40, purified water, sodium chloride, hydrochloric acid and/or sodium hydroxide to adjust the pH	solution/drops solution/drops	Fr USA	1996 2009
**Ketorolac Tromethamine 0.5%**		Benzalkonium chloride,			
Multidose bottle 3, 5 and 10 mL	Indomethacin Ketorolac tromethamine	Edetate disodium, octoxynol 40, sodium chloride, hydrochloric acid and/or sodium hydroxide to adjust the pH, purified water	solution/drops solution/drops	Fr USA	1997 2009
**Ketorolac Tromethamine 0.5%**		Benzalkonium chloride,			
3 mL in 5 mL multidose bottle 5 mL in 5 mL multidose bottle 10 mL in 10 mL multidose bottle	Ketorolac tromethamine Nepafenac	Edetate disodium, octoxynol 40, water for injection, sodium chloride, hydrochloric acid and/or sodium hydroxide to adjust the pH	solution/drops suspension/drops	USA EU	2009 2007
**Nevanac 0.1%**		Benzalkonium chloride,			
Multidose bottle 3 mL	Ketorolac tromethamine Nepafenac	Boric acid, propylene glycol, carbomer 974P, sodium chloride, guar gum, carboxymethylcellulose sodium, edetate disodium, hydrochloric acid and/or sodium hydroxide to adjust the pH, purified water	solution/drops suspension/drops	USA USA	2009 2005
**Nevanac 0.1%**		Benzalkonium chloride,			
3 mL in 4 mL multidose bottle	Ketorolac tromethamine Flurbiprofen sodium	Boric acid, propylene glycol, carbomer 974P, sodium chloride, tyloxapol, edetate disodium, hydrochloric acid and/or sodium hydroxide to adjust the pH, purified water	solution/drops solution/drops	USA Fr	2009 1991
**Ocufen 0.03%**					
Single use vial 0.4 mL	Ketorolac tromethamine Flurbiprofen sodium	Polyvinyl alcohol, sodium chloride, sodium citrate, potassium chloride, citric acid, hydrochloric acid and/or sodium hydroxide to adjust the pH, purified water	solution/drops solution/drops	USA USA	2018 1986
**Ocufen 0.03%**		Thimerosal,			
2.5 mL in 5 mL multidose bottle	Ketorolac tromethamine Bromfenac sodium	Citric acid, edetate disodium, polyvinyl alcohol, potassium chloride, purified water, sodium chloride, sodium citrate, hydrochloric acid and/or sodium hydroxide to adjust the pH	solution/drops solution/drops	USA USA	2009 2013
**Prolensa EQ 0.07% Acid**		Benzalkonium chloride,			
1.6 mL in 7.5 mL multidose bottle 13 mL in 7.5 mL multidose bottle	Ketorolac tromethamine Diclofenac sodium	Boric acid, edetate disodium, povidone, sodium borate, sodium sulfite, tyloxapol, sodium hydroxide, water for injection	solution/drops solution/drops	USA USA	2009 1991
**Voltaren 0.1%**					
Multidose bottle 5 mL	Ketorolac tromethamine Diclofenac sodium	Polyoxyl 35 castor oil, boric acid, tromethamine, sorbic acid, edetate disodium, purified water	solution/drops solution/drops	USA Fr	2009 1995
**Voltarenophta 0.1%**					
Single use vial 0.3 mL	Ketorolac tromethamine Diclofenac sodium	Cremophor EL, tromethamine, boric acid, water for injection	solution/drops solution/drops	USA Fr	2009 2005
**Voltarenophtabak 0.1%**					
Multidose bottle 10 mL	Nepafenac Bromfenac	Cremophor EL, tromethamine, boric acid, water for injection	suspension/drops solution/drops	EU EU	2007 2011
**Yellox 0.09%**		Benzalkonium chloride,			
Multidose bottle 5 mL	Nepafenac Bromfenac sodium	Boric acid, borax, sodium sulphite anhydrous (E221), tyloxapol, povidone, edetate disodium, water for injections, sodium hydroxide to adjust the pH	suspension/drops solution/drops	USA Fr	2005 2011
**Yellox 0.09%**		Benzalkonium chloride,			
Multidose bottle 5 mL	Flurbiprofen sodium	Boric acid, borax, sodium sulphite anhydrous (E221), tyloxapol, povidone, edetate disodium, water for injections, sodium hydroxide to adjust the pH	solution/drops	Fr	1991

**Table 6 pharmaceutics-12-00570-t006:** Topical ocular pharmaceutical forms and compositions of NSAIDs or SAIDs associated with anti-infective drugs in the US, European, or French markets listed as of 26th February, 2019.

Trade Name AND Presentation	Active Substance	SAID/ NSAID	Excipients	Pharmaceutical Form	Marketed In	Year of Authorization
**Bacitracin-Neomycin-Polymyxin W/ Hydrocortisone Acetate 400 UNITS/GM;1%;EQ 3.5MG BASE/Gm;10,000 Units/GM**	Hydrocortisone acetate, Bacitracin zinc, Neomycin sulfate, Polymyxin B sulfate	SAID		ointment	USA	1981
**-**						
**Blephamide 0.2%; 10%**	Prednisolone acetate, Sulfacetamide sodium	SAID	Benzalkonium chloride,	suspension/ drops	USA	1961
5 mL in 10 mL multidose bottle 10 mL in 15 mL multidose bottle			Edetate disodium, polysorbate 80, polyvinyl alcohol, potassium phosphate monobasic, purified water, sodium phosphate dibasic, sodium thiosulfate, hydrochloric acid and/or sodium hydroxide to adjust the pH			
**Blephamide S.O.P. 0.2%; 10%**	Prednisolone acetate, Sulfacetamide sodium	SAID	Phenylmercuric acetate,	ointment	USA	1986
3.5 g multidose tube			Mineral oil, petrolatum alcohol, lanolin alcohol, white petrolatum			
**Chibro Cadron 0.1%; 3 500 UNITS/ML**	Dexamethasone sodium phosphate, Neomycin sulfate	SAID	Benzododecinium bromide,	solution/drops	Fr	1992
Multidose bottle 5 mL			Sodium citrate, polysorbate 80, hydroxyethylcellulose, sodium hydroxide, sodium chloride, purified water, sodium citrate dihydrate			
**Cidermex 0.1%; 3 500 UNITS/GM**	Triamcinolone, Neomycin sulfate	SAID		ointment	Fr	1991
3 g multidose tube			Mineral oil, white petrolatum			
**Dexasporin 0.1%; EQ 3.5MG BASE/ML; 10 000 UNITS/ML**	Dexamethasone, Neomycin sulfate, Polymyxin B sulfate	SAID		suspension/ drops	USA	1995
-						
**Frakidex 0.1%; 6 300 UNITS/ML**	Dexamethasone sodium phosphate, Framycetine sulfate	SAID	Benzalkonium chloride,	solution/drops	Fr	1997
Multidose bottle 5 mL			sodium citrate, polysorbate 80, hydrochloric acid and/or sodium hydroxide to adjust the pH, purified water			
**Frakidex 0.1%; 3 150 UNITS/GM**	Dexamethasone sodium phosphate, Framycetine sulfate	SAID		ointment	Fr	1998
5 g multidose tube			Mineral oil, white petrolatum			
**Indobiotic 0.1%; 3 000 UNITS/ML**	Indomethacin, Gentamicin sulfate	NSAID		solution/drops	Fr	2000
Single use vial 0.35 mL			Hydroxypropylbetadex, arginine, hydrochloric acid, purified water			
**Maxidrol 0.1%; 3500 UNITS/ML; 6 000 UNITS/ML**	Dexamethasone, Neomycin sulfate, Polymyxin B sulfate	SAID	Benzalkonium chloride,	suspension/drops	Fr	1991
Multidose bottle 3 mL			Methylhydroxypropylcellulose, sodium chloride, polysorbate 20, hydrochloric acid and/or sodium hydroxide to adjust the pH, purified water			
**Maxidrol 0.1%; 3500 UNITS/GM; 6 000 UNITS/GM**	Dexamethasone, Neomycin sulfate, Polymyxin B sulfate	SAID	Methylparaben, propylparaben,	ointment	Fr	1997
3.5 g multidose tube			Lanolin, white petrolatum			
**Maxitrol 0.1%; EQ 3.5MG BASE/ML; 10 000 UNITS/ML**	Dexamethasone, Neomycin sulfate, Polymyxin B sulfate	SAID	Benzalkonium chloride,	suspension/ drops	USA	1963
5 mL in 8 mL multidose bottle			Hypromellose, sodium chloride, polysorbate 80, hydrochloric acid and/or sodium hydroxide to adjust the pH, purified water			
**Maxitrol 0.1%; EQ 3.5MG BASE/GM; 10 000 UNITS/GM**	Dexamethasone, Neomycin sulfate, Polymyxin B sulfate	SAID	Methylparaben, propylparaben,	ointment	USA	1963
3.5 g multidose tube			White petrolatum, anhydrous liquid lanolin			
**Maxitrol 0.1%; EQ 3.5MG BASE/ML; 10 000 UNITS/ML**	Dexamethasone, Neomycin sulfate, Polymyxin B sulfate	SAID		suspension/ drops	USA	1984
-						
**Neomycin AND Polymyxin B Sulfates AND Dexamethasone 0.1%; EQ 3.5MG BASE/GM; 10 000 UNITS/GM**	Dexamethasone, Neomycin sulfate, Polymyxin B sulfate	SAID	Methylparaben, propylparaben,	ointment	USA	1994
3.5 g multidose tube			White petrolatum, lanolin, mineral oil			
**Neomycin AND Polymyxin B Sulfates AND Dexamethasone 0.1%; EQ 3.5MG Base/GM; 10 000 UNITS/GM**	Dexamethasone, Neomycin sulfate, Polymyxin B sulfate	SAID		ointment	USA	1989
-						
**Neomycin AND Polymyxin B Sulfates AND Hydrocortisone 1%; EQ 3.5MG BASE/ML; 10 000 UNITS/ML**	Hydrocortisone, Neomycin sulfate, Polymyxin B sulfate	SAID	Potassium metabisulfite,	suspension/ drops	USA	1988
Multidose bottle 10 mL			Glycerin, propylene glycol, hydrochloric acid, water for injection			
**Neomycin AND Polymyxin B Sulfates, Bacitracin ZINC AND Hydrocortisone 400 UNITS/GM; 1%; EQ 3.5MG BASE/GM; 10 000 UNITS/GM**	Hydrocortisone acetate, Bacitracin zinc, Neomycin sulfate, Polymyxin B sulfate	SAID		ointment	USA	1995
3.5 g multidose tube			Mineral oil, white petrolatum			
**Neomycin AND Polymyxin B Sulfates, Bacitracin Zinc AND Hydrocortisone 400 UNITS/GM; 1%; EQ 3.5MG BASE/GM; 10 000 UNITS/GM**	Hydrocortisone acetate, Bacitracin zinc, Polymyxin B sulfate Neomycin sulfate,	SAID		ointment	USA	2012
3.5 g multidose tube						
**Pred-G EQ 0.3%; 0.6%**	Prednisolone acetate, Gentamicin sulfate	SAID	Chlorobutanol,	ointment	USA	1989
3.5 g multidose tube			Mineral oil, petrolatum, lanolin alcohol, white petrolatum			
**Pred-G EQ 0.3%; 1%**	Prednisolone acetate, Gentamicin sulfate	SAID	Benzalkonium chloride,	suspension/ drops	USA	1988
5 ml in 10 mL multidose bottle 10 ml in 15 mL multidose bottle			Edetate disodium, hypromellose, polyvinyl alcohol, polysorbate 80, purified water, sodium chloride, sodium citrate dihydrate, hydrochloric acid and/or sodium hydroxide to adjust the pH			
**Sterdex**	Dexamethasone, Axytetracycline	SAID		ointment	Fr	1997
Single dose vial			Mineral oil, white petrolatum			
**Tobradex 0.1%; 0.3%**	Dexamethasone, Tobramycin	SAID	Benzalkonium chloride,	suspension/ drops	USA	1988
10 ml in 15 mL multidose bottle			Tyloxapol, edetate disodium, sodium chloride, hydroxyethyl cellulose, sodium sulfate, sulfuric acid and/or sodium hydroxide to adjust the pH, purified water			
**Tobradex 0.1%; 0.3%**	Dexamethasone, Tobramycin	SAID	Chlorobutanol,	ointment	USA	1988
3.5 g multidose tube			Mineral oil, white petrolatum			
**Tobradex ST 0.05%; 0.3%**	Dexamethasone, Tobramycin	SAID	Benzalkonium chloride,	suspension/ drops	USA	2009
Multidose bottle 2.5, 5 and 10 mL			Xanthan gum, tyloxapol, edetate disodium, sodium chloride, propylene glycol, sodium sulfate, hydrochloric acid and/or sodium hydroxide to adjust the pH, purified water			
**Tobradex 0.1%; 0.3%**	Dexamethasone, Tobramycin	SAID	Benzalkonium chloride,	suspension/ drops	Fr	1997
Multidose bottle 5 mL			Edetate disodium, sodium chloride, sodium sulfate, tyloxapol, hydroxyethylcellulose, sulfuric acid and/or sodium hydroxide to adjust the pH, purified water			
**Tobramycin AND Dexamethasone 0.1%; 0.3%**	Dexamethasone, Tobramycin	SAID	Benzalkonium chloride,	suspension/ drops	USA	1991
Multidose bottle 2.5 and 5 mL			Sodium sulfate, sodium chloride, hydroxyethylcellulose, tyloxapol, edetate disodium, purified water, sulfuric acid and/or sodium hydroxide to adjust the pH			
**Zylet 0.5%; 0.3%**	Loteprednol etabonate, Tobramycin	SAID	Benzalkonium chloride,	suspension/ drops	USA	2004
5 mL in 7.5 mL multidose bottle 10 mL in 10 mL bottle			Edetate disodium, glycerin, povidone, purified water, tyloxapol, sulfuric acid and/or sodium hydroxide to adjust the pH			
**Prednisolone Sodium Phosphate EQ 0.23%; Sulfacetamide Sodium 10%**	Prednisolone sodium phosphate, Sulfacetamide sodium	SAID		solution/drops	USA	1993
-						
**Prednisolone Sodium Phosphate EQ 0.23%; Sulfacetamide Sodium 10%**	Prednisolone sodium phosphate, Sulfacetamide sodium	SAID	Thimerosal,	solution/drops	USA	1995
Multidose bottle 5 and 10 mL			Poloxamer 407, boric acid, edetate disodium, purified water, hydrochloric acid and/or sodium hydroxide to adjust the pH			

**Table 7 pharmaceutics-12-00570-t007:** Topical ocular pharmaceutical forms and compositions of ciclosporin in the US, European, or French markets listed as of 26th February, 2019.

Trade Name AND Presentation	Excipients	Pharmaceutical Form	Marketed In	Year of Authorization
**Cequa 0.09%**		solution/drops	USA	2018
Single use vial 0.25 mL	Polyoxyl-35 castor oil, octoxynol 40, polyvinylpyrrolidone, sodium phosphate monobasic, sodium phosphate dibasic, hydrochloric acid and/or sodium hydroxide to adjust the pH, water for injection			
**Ikervis 1 mg/mL**		emulsion	EU, Fr	2015
Single use vial 0.3 mL	Medium-chain triglycerides, cetalkonium chloride, glycerol, tyloxapol, poloxamer 407, sodium hydroxide to adjust the pH, water for injection			
**Restasis 0.05%**		emulsion	USA	2002
Single use vial 0.4 mL	Glycerin, castor oil, polysorbate 80, carbomer 1342, sodium hydroxide to adjust the pH, purified water			
**Restasis 0.05%**		emulsion	USA	2016
5.5 mL in 10 mL multidose bottle	Glycerin, castor oil, polysorbate 80, carbomer copolymer A, sodium hydroxide to adjust the pH, purified water			
**Verkazia 1mg/mL**		emulsion	EU	2018
Single use vial 0.3 mL	Medium-chain triglycerides, cetalkonium chloride, glycerol, tyloxapol, poloxamer 407, sodium hydroxide to adjust the pH, water for injection			

**Table 8 pharmaceutics-12-00570-t008:** NSAIDs formulated in micro or nanocarriers for topical ophthalmic administration and their main components from the literature.

Drug	System	Main Components	Key Results	Ref.
**Aceclofenac**	Nanoparticles	EUDRAGIT^®^RS 100, Polysorbate 80, mannitol, water	High entrapment efficiency (>90%) Sustained drug release *in vitro* 2-fold higher transcorneal permeation *ex vivo* as compared with aceclofenac solution Higher anti-inflammatory activity *in vivo* than marketed formulation	[100]
		EUDRAGIT^®^RL 100, Polysorbate 80, mannitol, water	High entrapment efficiency (>95%) 2-fold higher transcorneal permeation *ex vivo* as compared with aceclofenac solution Higher anti-inflammatory activity *in vivo* than marketed formulation	[101]
**Amfenac**	Nanoparticles	Catechin, HAuCl4, tris acetate buffer, water	No irritation effect *in vivo* and no cytotoxic effect *in vitro* Higher efficiency in DED treatment *in vivo* than marketed formulation of ciclosporin A	[102]
**Bromfenac Sodium**	Liposomes	L-α-distearoylphosphatidylcholine, dicetylphosphate, cholesterol, acetate salt solution, Hank’s balanced salt solution, 2-morpholinoethanesulfonic acid monohydrate, chitosan, water	Good entrapment efficiency (>75%) Sustained drug release without burst effect *in vitro*	[103]
**Celecoxib**	Nanoparticles	Poly-ε-caprolactone, poloxamer 188, Sorenson’s phosphate buffer, water	High entrapment efficiency (>89%) Sustained drug release without burst effect *in vitro* ≈ 2-fold higher corneal permeation *ex vivo* Higher anti-inflammatory activity *in vivo* than marketed formulation	[104]
	Solid lipid nanoparticles	Lipid glyceryl monostearate, PVA, polysorbate 80, poloxamer 188, Sorenson’s phosphate buffer, water	Entrapment efficiency (65 < X < 94%) Sustained drug release with burst effect *in vitro* ≈ 2-fold higher corneal permeation *ex vivo* Higher anti-inflammatory activity *in vivo* than marketed formulation	[104]
**Dexibuprofen**	Nanoparticles	PLGA-PEG 5%, PVA, water	No irritant effect *in vitro* and *in vivo* High entrapment efficiency (>85%) Sustained drug release up to 12 h *in vitro* and *ex vivo* Sustained anti-inflammatory activity *in vivo*	[105]
**Diclofenac**	Nanoparticles	Methoxy poly(ethylene glycol)-poly(ε-caprolactone)-chitosan copolymer, sodium chloride, water	No cytotoxic effect *in vitro* no irritation effect *in vivo* High entrapment efficiency (>95%) Sustained drug release up to 8 h *in vitro* ≈1.4-fold higher corneal penetration *ex vivo* than marketed formulation 2.3-fold higher concentration in aqueous humor *in vivo* than marketed formulation	[106]
		NaOH, Zn(NO_3_)_2_ 6H_2_O, Al(NO_3_)_3_ 9H_2_O, PVP K30, trichlorobutanol, water	No irritation effect *in vivo* High corneal penetration *ex vivo* High apparent permeability coefficient and prolonged precorneal retention time *in vivo*	[107]
**Diclofenac Sodium**	Liposomes	Phosphatidylcholine, cholesterol, phosphatidylserine low molecular weight chitosan and sodium chloride, water	No irritation effect *in vivo* High entrapment efficiency (>95%) ≈2-fold higher corneal penetration at 6 h *ex vivo* than diclofenac solution	[108]
	Micelles	Methoxypoly(ethylene glycol)-poly(ε-caprolactone), water	No irritation effect *in vivo* Good entrapment efficiency (>70%) Sustained drug release *in vitro* up to 24 h 17-fold higher corneal penetration *ex vivo* 3-fold higher concentration in aqueous humor *in vivo* 2-fold higher bioavailability *in vivo*	[109]
**Diclofenac Sodium**	Nanoparticles	N-trimethyl chitosan, phosphate buffer, polysorbate 80, sodium tripolyphosphate, water	No irritating effect *in vitro* and *in vivo* Entrapment efficiency >70% Sustained drug release *in vitro* ≈2-fold higher concentration in aqueous humor *in vivo* at 1 h	[95]
	Nanoparticles	PLGA, poly[Lac(Glc-Leu)], polysorbate 80, benzalkonium chloride, mannitol, water	No irritants effect *in vivo* Sustained drug release *in vitro* up to 14 h	[110]
	Solid lipid nanoparticles	PHOSPHOLIPON 90G^®^, goat fat, polysorbate 80, sorbitol, thimerosal, water	High entrapment efficiency (≈90%) Sustained drug release *in vitro* Higher corneal permeation flux	[111]
**Flurbiprofen**	Cubosomes	Glyceryl monooleate, poloxamer 407, glycerol, water	No irritation effect *in vivo* High entrapment efficiency (>98%) Sustained drug release without burst effect *in vitro* 2.5- and 2-fold higher apparent permeability *ex vivo* 2-fold higher aqueous humor concentration *in vivo* at 3 h	[112]
	Liposomes	Chitosan, egg phosphatidylcholine, cholesterol, SOLUTOL^®^HS-15, HCl, water	No irritation effect *in vivo* High encapsulation efficiency (>90%) 4.59-, 3.56- and 2.36-fold higher apparent permeability *ex vivo* 4.11- and 2.19-fold higher prolonged retention time *in vivo*	[113]
	Nanoemulsion	PLGA, poloxamer 188, water	High entrapment efficiency (>85%) Sustained drug release *in vitro* ≈1.7-fold increase corneal permeation *ex vivo* than marketed formulation	[114]
	Nanoparticles	EUDRAGIT^®^RS 100 and RL 100, polysorbate 80, phosphate buffer, benzalkonium chloride, water	No irritation effect *in vivo* High entrapment efficiency (>85%) Sustained drug release without burst effect Higher concentration in aqueous humor than with marketed formulation	[115]
		PLGA, poloxamer 188, PVA, water	No irritation effect *in vivo* Good entrapment efficiency (>75%) Sustained drug release *in vitro* Higher anti-inflammatory activity *in vivo* than marketed formulation	[116]
		Poly-ε-caprolactone, poloxamer 188, water	Entrapment capacity (>75%) Sustained drug release *in vitro*	[117,118]
		PLGA or poly-ε-caprolactone, water	Good entrapment efficiency (>85%) ≈ 3.9- and 7.6-fold increase corneal permeation *ex vivo*	[119]
		PLGA, poloxamer 188, water	No irritation effect *in vitro* High entrapment efficiency (>90%) Sustained drug release *in vitro*	[120]
		Poly-ε-caprolactone, poloxamer 188, trehalose or PEG3350, water	No irritating effect *in vitro* and *in vivo* Good entrapment efficiency (>85%) Sustained drug release *in vitro* Enhance corneal permeation *ex vivo* Higher anti-inflammatory activity *in vivo*	[121]
	Solid lipid nanoparticles	Stearic acid, MIGLYOL^®^ 812, castor oil, polysorbate 80, water	No irritation effect *in vivo* Good entrapment efficiency (>75%) Sustained drug release without burst effect *in vitro*	[122]
**Flurbiprofen Axetil**	Nanoemulsion	Castor oil, polysorbate 80, glycerin, carbomer 974P, sodium acetate, boric acid, sorbic acid, water	High entrapment efficiency (>98%) Better ocular biocompatibility than marketed formulation Higher anti-inflammatory activity *in vivo* than marketed formulation	[123]
**Ibuprofen**	Liposomes	Soybean phospholipids, cholesterol, octadecylamine, water	72.9 % entrapment efficiency 1.64-fold higher corneal permeation *ex vivo* at 6 h 1.53-fold higher aqueous humor concentration *in vivo*	[124]
	Liposomes	Cotton-like silk fibroin, phosphate buffer, purified soybean lecithin, cholesterol, stearylamine, water	No cytotoxic effect *in vitro* Entrapment efficacy (59 < X < 86%) Sustained release *in vitro* and sustained corneal permeation *ex vivo*	[125]
	Solid lipid nanoparticles	Polyoxyl-35 castor oil, COMPRITOL^®^ 888 ATO, Gelucire 44/14 or TRANSCUTOL^®^ P or stearylamine, MIGLYOL^®^ 812, water	High entrapment efficiency (>90%) 4.19-fold higher corneal apparent permeability *ex vivo* 3.99-fold increase of aqueous humor drug concentration *in vivo*	[126]
**Ibuprofen Sodium Salt**	Nanoparticles	EUDRAGIT^®^RS 100, polysorbate 80, water	Higher anti-inflammatory activity *in vivo* than marketed formulation ≈ 1,5-fold higher aqueous humor concentration *in vivo* than with ibuprofen solution	[127]
	Nanoparticles	EUDRAGIT^®^ RS 100, polysorbate 80, benzalkonium chloride, water	Good ocular tolerability High entrapment efficiency (>90%) Sustained drug release Higher aqueous humor concentration *in vivo*	[128]
**Indomethacin**	Microparticles/ Nanoparticles	Zirconia beads and Bead Smash 12, benzalkonium chloride, mannitol or methylcellulose, HPβCD, sodium chloride, water	Better ocular tolerance than marketed formulation *in vitro* ≈ 6-fold higher corneal penetration *in vitro* ≈ 10-fold higher corneal penetration *in vivo*	[129]
	Nanoemulsion/ Nanoparticles	NC: Poly-ε-caprolactone, lecithin, MIGLYOL^®^ 840, poloxamer 188, water NE: Lecithin, MIGLYOL^®^ 840, poloxamer 188, water NP: Poloxamer 188, water	Good tolerance *in vivo* High entrapment efficiency (>89%) Sustained drug release 4–5-fold higher corneal penetration *ex vivo* than marketed formulation	[130]
	Nanoemulsion/Nanoparticles	NP: Chitosan with tripolyphosphate, acid acetic, water NE: Chitosan, lecithin soya, MIGLYOL^®^ 840 and Poloxamer 188 or PVA or polysorbate 80, sorbitol, benzalkonium chloride, water	Good entrapment efficiency (>75%) Sustained release *in vitro* 30-fold higher corneal concentration *in vivo* at 1 h with NE than with solution 13-fold higher aqueous humor *in vivo* at 6 h post instillation with NE than with solution	[97]
	Nanoparticles	Poly-ε-caprolactone, lecithin, MIGLYOL^®^ 840, poloxamer 188, poly-l-lysin or chitosan, water	Good tolerance *in vivo* High entrapment efficiency (>90%) Rapid release *in vitro* 4-6 and 4-7-fold higher corneal and aqueous humor concentrations *in vivo* after 30 and 60 minS post-instillation than marketed formulation	[131]
	Solid lipid nanoparticles	COMPRITOL^®^ 888 ATO, poloxamer 188 and/or polysorbate 80, glycerin, NaOH or HCl, water	Entrapment efficiency (>70%) 3 – 4.5-fold higher corneal permeability *ex vivo* than marketed formulation	[132]
**Ketorolac Tromethamine**	Micelles	*Copolymer of N*-isopropylacrylamide, vinyl pyrrolidone and acrylic acid crosslinked with *N*,*N*′-methylene *bis*-acrylamide, water	30% entrapment efficiency Sustained release *in vitro* 2-fold higher corneal permeation *ex vivo* Higher anti-inflammatory activity up to 3 h and PMN migration *in vivo*	[94]
	Nanoparticles	Chitosan, acetic acid, NaOH, tripolyphosphate, water	Entrapment efficiency (34 < X < 41%) Sustained drug release	[133]
		Chitosan, acetic acid, tripolyphosphate, NaOH, water	Entrapment efficiency (5 < X < 75%) Sustained release *in vitro* up to 6 h 3.77-fold lower permeation parameters lower than solution *ex vivo*	[134]
**Naproxen**	Microparticles	Sodium alginate, carbomer 974P, hydroxypropyl methylcellulose, paraffin, calcium chloride, water	Good entrapment efficiency (63 < X < 76%) Sustained release *in vitro* without burst effect	[135]
	Nanoparticles	PLGA, PVA, water	High entrapment efficiency (>80%) Sustained drug release *in vivo* without burst effect *in vitro*	[136]
**Nepanefac**	Nanoaggregates	PVP, PVA, carboxymethylcellulose, hydroxypropylmethylcellulose, methyl cellulose, tyloxapol, γCD, HPβCD, EDTA, benzalkonium chloride, sodium chloride, water	Good entrapment efficiency (>60%)	[137]
**Phospho-Sulindac**	Nanoparticles	Methoxy poly(ethylene glycol)-poly(lactide), sodium cholate, water, phosphate buffer	Entrapment efficacy 46.4% Sustained drug release *in vitro* up to 24 h	[138]
**Piroxicam**	Microparticles	Albumin, sodium chloride or sorbitol, water	High entrapment efficiency (>99%) Sustained release *in vitro* 1.8-fold higher bioavailability *in vivo* than marketed formulation	[139]
	Nanoparticles	EUDRAGIT^®^RS 100, hydroxypropyl methylcellulose, PVA, sodium chloride, water	Sustained release *in vitro* Great anti-inflammatory activity *in vivo* up to 12 h but no difference compared with microsuspension	[140]

α-CD: α-cyclodextrin, βCD: β-cyclodextrin, γCD: γ-cyclodextrin, HPβCD: hydroxypropyl-β-cyclodextrin, HPγCD: hydroxypropyl- γ-cyclodextrin, RMβCD: randomly methylated-β-cyclodextrin, PEG: polyethylene glycol, PLGA: poly(lactic-*co*-glycolic acid), Poly[Lac(Glc-Leu)]: poly(lactide-*co*-glycolide-leucine), PVA: polyvinyl alcohol, PVP: polyvinylpyrrolidone, EDTA: ethylenediaminetetraacetic acid, HCl: hydrochloric acid, NaOH: sodium hydroxide.

**Table 9 pharmaceutics-12-00570-t009:** SAIDs and SAIDs associated with anti-infective formulated in micro or nanocarriers for topical ophthalmic administration.

Drug	System	Main Components	Key Results	Ref.
**Dexamethasone**	Cubosomes	Monoolein, poloxamer 407, glycerol, water	Good tolerance *in vitro* High entrapment efficiency (>95%) 4.5 - 3.5-fold higher apparent permeability *in vitro* 1.8 fold increase the concentration in aqueous humor *in vivo*	[141]
	Microemulsion	Isopropyl myristate, polysorbate 80, propylene glycol, chitosan, acetate buffer, water	No irritation effect *in vivo* High entrapment efficiency (>95%) Sustained drug release with burst effect *in vitro* Higher anti-inflammatory activity *in vivo* than marketed formulation	[142]
	Microparticles/Nanoparticles	Zirconia beads and Bead Smash 12,methylcellulose, propyl p-hydroxybenzoate, methyl p-hydroxybenzoate, water	No cytotoxic effect *in vitro* ≈ 5.1-fold higher corneal penetration of nanoparticles than marketed formulation *in vivo*	[143]
	Nanogels suspension	HPγCD, γCD nanogels, EDTA, benzalkonium chloride, hydroxypropylmethylcellulose, sodium chloride, pH adjuster, water	No irritation effect *in vitro* and *in vivo* Sustained drug release without burst effect ≈ 80-fold increase concentration in tear fluid at 6 h *in vivo* 3-fold increase concentration in aqueous humor *in vivo*, 2 h after instillation	[144]
		*N*-tert-butylacrylamide, methylcellulose, nitric acid, cerium ammonium nitrate, water	No cytotoxic effect *in vitro* High entrapment encapsulation efficiency (>95%) Sustained drug release without burst effect	[145]
	Nanomicelles	Polyoxyl-40-stearate, polysorbate 80, water	No irritation effect *in vivo* Sustained drug release *in vitro*	[146]
	Nanoparticles	Ethyl cellulose or EUDRAGIT^®^ RS or ethyl cellulose/EUDRAGIT^®^ RS, PVA, water	No toxicity, except for ethylcellulose particles Entrapment efficiency (12 < X < 87%) Sustained drug release without burst release	[147]
		Propylene glycol, phosphate buffer, EDTA, poloxamer 188, hydroxyethylcellulose, benzalkonium chloride, water	Higher intensity of drug action Higher extent of drug absorption	[148]
		γCD, HPγCD, poloxamer 407, benzalkonium chloride, EDTA, sodium chloride, water	15-fold higher concentration than marketed formulation	[88]
	Nanosponges	βCD nanosponge, water	No irritation or toxic effect *ex vivo* Entrapment efficiency (3 < X < 10%) Sustained drug release without burst effect ≈2-fold higher corneal permeability *ex vivo*	[149]
	Solid lipid nanoparticles	Soy lecithin, soybean oil, glycerol, poloxamer 188+/-chitosan, water	No irritation effect *in vivo* Entrapment efficiency (30 < X < 70%) Sustained drug release *in vitro* 4.69-fold higher concentration in aqueous humor from L/NPs with chitosan than aqueous solution *in vivo*	[150]
**Dexamethasone Sodium Phosphate**	Microparticles	RMβCD or γCD, benzalkonium chloride, EDTA, sodium chloride, hydroxypropylmethylcellulose, water	No irritation effect *in vivo* 3-8-fold higher concentration in aqueous humor 2 h after instillation *in vivo* than marketed formulation	[151]
	Nanoparticles	Chitosan, sodium tripolyphosphate, acid acetic, phosphate buffer, hyaluronic acid, water	No irritation effect *in vivo* Entrapment efficiency (58 < X < 73%) Sustained drug release *in vitro* Prolonged precorneal retention *in vivo* ≈ 8-fold increase the aqueous concentration at 6 h *in vivo*	[152,153]
		Quaternary ammonium-chitosan conjugate or its thiolated derivative, acid hyaluronic, phosphate buffer, water	No irritation effect *in vivo* Entrapment efficiency (18 < X < 35%) Sustained drug release *in vitro* Sustained residence time in tear fluid *in vivo*	[154]
**Fluocinolone Acetonide**	Liposomes	α-, β and HPβCD, water, dextrose, glucose, phosphatidyl choline, triolein, cholesterol, L-lysine, phosphate buffer, water	Entrapment efficiency (7 < X < 52%) Sustained release *in vitro* up to 180h for FA-HPβCD complex	[155]
	Nanoparticles	PLGA P 5002 or 7502, poloxamer 407, phosphate buffer, chitosan HCl, water	No irritation effect *in vivo* Entrapment efficiency (> 50%) Sustained drug release *in vitro* ≈ 2.5-fold higher concentration in tear sample *in vivo* at 1h	[156]
**Fluoro-Metholone**	Nanoparticles	PLGA, poloxamer188, water	No irritation effect *in vitro* and *in vivo* High entrapment efficiency (>99%) Sustained drug release *in vitro* ≈2.2-fold higher increase corneal permeation *ex vivo* than marketed formulation Higher anti-inflammatory activity *in vivo* at 30 mins than marketed formulation	[91]
**Hydrocortisone**	Micelles/Nanoparticles	Albumin, glutaraldehyde, sodium metabisulfite, glucose, polysorbate 80, phosphate buffer, water	Entrapment efficiency (16 < X < 70%) Sustained corneal permeation *ex vivo* Neither higher AUC values nor prolonged release *in vivo*	[157]
	Nanoparticles	Propylene glycol, isotonic phosphate buffer, EDTA, hydroxyethylcellulose, benzalkonium chloride, poloxamer 188, water	Higher intensity of drug action Higher extent of drug absorption	[148]
		Gelatin A or B, water, HCl or NaOH, sodium metabisulfite, HPβCD, glutaraldehyde, water	Entrapment efficiency (35 < X < 45%) Sustained drug release *in vitro* closed to zero order, 30% in 200 min	[158]
**Loteprednol Etabonate**	Nanogels suspension	*N*-boc ethylenediamine, polysorbate 60, chitosan, succinic anhydride, 1-ethyl-3-(3-dimethylaminopropyl) carbodiimide, *N*-hydroxysuccinimide, phosphate buffer, water	No cytotoxic effect *in vitro* Good entrapment efficiency (67 < X < 70%) Sustained release *in vitro*	[159]
	Nanoparticles	PLGA, PVA, water	Good entrapment efficiency (>70%) Improve *ex vivo* transcorneal penetration	[160]
**Methyl-Prednisolone Acetate**	Nanoparticles	EUDRAGIT^®^RS 100, PVA, sodium chloride, hydroxypropylmethylcellulose, water	No irritation effect *in vivo* Sustained release *in vitro* Higher anti-inflammatory activity up to 36 h *in vivo*	[161]
**Pirfenidone**	Nanoparticles	Monoolein, poloxamer P 407, oleic acid, NaOH, glycerin, water	No irritation effect *in vitro* Entrapment efficiency (6 < X < 36%) Sustained release *in vitro* Reduction in ocular lesions associated with a reduction of inflammatory cells *in vivo*	[162]
**Prednisolone**	Micelles	Quaternary ammonium palmitoyl gycol chitosan, poloxamer 407, water	45% entrapment efficiency 10-fold aqueous humor concentration *in vivo*	[163]
	Nanoparticles	Poly-ε-caprolactone or EUDRAGIT^®^ RS100, castor oil and mineral oil, sorbitan monostrearate, polysorbate 80, water	No irritation effect *in vitro*, no cytotoxic effect *in vitro* Entrapment efficiency (45 < X < 52%) Sustained release *in vitro*	[164]
		Propylene glycol, phosphate buffer, EDTA, hydroxyethylcellulose, benzalkonium chloride, poloxamer 188, water	Higher intensity of drug action Higher extent of drug absorption	[148]
**Prednisolone Acetate**	Liposomes	1,2-dipalmitoyl-sn-glycerol-3-phosphocholine, cholesterol, stearylamine, water	High entrapment efficiency (78 < X < 90%) Sustained release *in vitro* 1.2 – 2.8-fold lower apparent corneal permeability *ex vivo* than solution ≈ 3 – 5-fold higher aqueous humor concentration at 3 h *in vivo* than solution Higher anti-inflammatory activity *in vivo* with positively charged unilamelar liposome	[165]
**Prednisolone Acetate or Phosphate**	Ethoniosomes	SPAN^®^ 60, cholesterol, phosphate buffer, water	No irritation effect *in vivo* Entrapment efficiency >85% for prednisolone acetate and 25 < X < 46% for Prednisolone phosphate Sustained release *in vitro* Higher corneal permeation than marketed formulation Lower bioavailability than marketed formulation Quicker anti-inflammatory activity than marketed formulation	[166]
**Prednisolone Gatifloxacine**	Nanoparticles	EUDRAGIT^®^RS 100, RL 100, hyaluronic acid, benzalkonium chloride, EDTA, water	Good entrapment efficiency (>60%) Sustained release *in vitro* 5.23-fold higher and sustained concentration in aqueous humor *in vivo* than marketed formulation	[167]
**Triamcinolone Acetonide**	Nanoparticles	Methoxypoly(ethylene glycol)-poly(dl-lactic-*co*-glycolic acid),PVA, water	No cytotoxic effect *in vitro* 77% entrapment efficiency Sustained release maintained for 45 days *in vitro* anti-inflammatory activity *in vivo*	[168]
		Poly-ε-caprolactone, poloxamer 188, water	No cytotoxic effect *in vitro* 60% encapsulation efficiency Sustained release *in vitro* anti-inflammatory activity *in vivo*	[169]
		PLGA, PVA, water	Poor entrapment efficiency (12 < X < 32%) Sustained release *in vitro* Similar anti-inflammatory activity *in vivo* than intravitreal injection	[170]

α-CD: α-cyclodextrin, βCD: β-cyclodextrin, γCD: γ-cyclodextrin, HPβCD: hydroxypropyl-β-cyclodextrin, HPγCD: hydroxypropyl- γ-cyclodextrin, RMβCD: randomly methylated-β-cyclodextrin, PLGA: poly(lactic-*co*-glycolic acid), PVA: polyvinyl alcohol, EDTA: ethylenediaminetetraacetic acid, HCl: hydrochloric acid, NaOH: sodium hydroxide.

**Table 10 pharmaceutics-12-00570-t010:** NSAIDs formulated in combined strategies for topical ophthalmic administration.

Drug	System	Main Components	Key Results	Ref.
**Celecoxib**	Nanoparticles in gel	Lecithin, poloxamer 188, PVA, poly-ε-caprolactone or PLA or PLGA, trehalose, hydroxypropylmethylcellulose or methylcellulose, phosphate buffer, benzalkonium chloride, water	No cytotoxic effect *in vitro* Good entrapment efficiency (>79%) Sustained drug release without burst effect *in vitro*	[176]
**Celecoxib**	Nanoparticles in gel	Chitosan or sodium alginate, poly-ε-caprolactone or PLA or PLGA, lecithin, PVA, poloxamer 188, trehalose, hydroxypropylmethylcellulose or methylcellulose, phosphate buffer, benzalkonium chloride, water	≈ 5-fold higher concentration in aqueous humor *in vivo* 4.8–29.7-fold higher bioavailability *in vivo* than marketed formulation	[174]
**Celecoxib**	Nanoparticles in gel	Chitosan or poly-ε-caprolactone, sodium alginate, lecithin, PVA or poloxamer 188, acetic solution, trehalose, hydroxypropylmethylcellulose or methylcellulose, benzalkonium chloride, water	No cytotoxic effects *in vitro* Entrapment efficiency (>75%) Sustained drug release without burst effect *in vitro*	[177]
**Diclofenac**	Micelles in gel	Methoxypoly(ethylene glycol)-poly-ε-caprolactone copolymer,αCD, water	Low cytotoxic effects *in vitro* No irritant effects *in vivo* Sustained drug release *in vitro* up to 216 h, 2.37-fold higher concentration in aqueous humor *in vivo* 1h after instillation compared to micelles	[178]
**Flurbiprofen**	Solid lipid nanoparticles in gel	COMPRITOL^®^ 888 ATO, saturated fatty acid of C18, Gelificante PFC carbomer, MIGLYOL^®^ 812, castor oil, Polysorbate 80, glycerol, NaOH, water	No irritation effects *in vivo* Good entrapment efficiency (>70%) Sustained release without burst effect *in vitro*, Higher corneal permeation *ex vivo*	[179]
**Ibuprofen**	Solid lipid nanoparticles in-situ forming gel	COMPRITOL^®^ 888 ATO, MIGLYOL^®^ 812, cetyltrimethylammonium bromide, Polysorbate 80, poloxamer 407, water	No cytotoxic *in vitro* High entrapment efficiency (>90%) Sustained release *in vitro*	[180]
**Ketorolac Tromethamine**	Nanoparticles in-situ forming gel	EUDRAGIT^®^RL 100, poloxamer 407, hydroxypropylmethylcellulose, citrate-phosphate buffer, PVA, water	No irritation effect *in vivo* Entrapment efficiency (51 < X < 92%) ≈ 3-fold higher corneal permeation *ex vivo* ≈ 4-fold higher concentration in aqueous humor *in vivo* at 4 h	[181]
**Meloxicam**	Nanoaggregates in contact lens	Bovine serum albumin, polysorbate 80, NaOH, HCl, 2-HEMA monomer, tetraethylene glycol dimethacrylate, ethylene glycol, sodium metabisulfite, ammonium persulfate, water	No irritation effect *in vivo* Sustained drug release without burst effect *in vitro* Reduce corneal penetration *ex vivo*	[182]
**Nepanefac**	Nanoparticles in-situ forming gel	Tetraethyl orthosilicate, cetyltrimethyl ammonium bromide, ammonia, polysorbate 80, poloxamer 407, Pluronic F67 or chitosan, water	No cytotoxic effect *in vitro* High entrapment capacity (>98%) Sustained drug release *in vitro* 3.68-fold higher corneal permeation *ex vivo*	[183]
**Piroxicam**	Microparticles /Microparticles in gel	Pectine, polyacrylate gel, water	Entrapment efficiency (41 < X < 46%) ≈ 5–6-fold higher residence time *in vivo* ≈ fold increase bioavailability in aqueous humor *in vivo* than marketed formulation	[184]
**Pranoprofen**	Nanoparticles in gel	PLGA, PVA, carbomer 934P, glycerol, glycerin or azone, water	No irritation effects *in vitro* and *in vivo* High entrapment efficiency (>80%) Sustained release *in vitro* Greater anti-inflammatory effect in the cornea *in vivo* than marketed formulation	[185]

α-CD: α-cyclodextrin, PLA: polylactic acid), PLGA: poly(lactic-co-glycolic acid), PVA: polyvinyl alcohol, 2-HEMA monomer: 2-Hydroxyethyl methacrylate monomer, HCl: hydrochloric acid, NaOH: sodium hydroxide

**Table 11 pharmaceutics-12-00570-t011:** SAIDs formulated in combined strategies for topical ophthalmic administration.

Drug	System	Main Components	Key Results	Ref.
Dexamethasone	Nanoparticles in-situ forming gel	Poloxamer 188, poloxamer 407, water	No irritation effects *in vivo* Sustained drug release *in vitro* 2.56-fold higher corneal permeation *ex vivo* ≈3-fold higher concentration in aqueous humor *in vivo*	[186]
Dexamethasone	Solid lipid nanoparticles in gel	Soybean oil, glycerol, poloxamer 188, poloxamer 407, water	No irritation effects *in vivo* Entrapment efficiency >50% Sustained drug release 2.56-fold increase corneal permeability ex vitro ≈3-fold higher concentration in aqueous humor *in vivo* at 6h after instillation than marketed formulation	[187]
Dexamethasone Acetate	Nanoparticles in film hydrogel	Kaolin, hydroxypropyl methylcellulose 5 and 15000cps, triethanolamine, water	Poor entrapment efficiency (8.89–9.8%) Controlled drug releases *in vitro* up to 6h without burst effect Kaolin extends the corneal permeation up to 6h *ex vivo* Sustained anti-inflammatory activity *in vivo*	[188]
Fluorometholone	Nanoparticles in-situ forming gel	PLGA, poloxamer 407, sodium alginate, sodium carboxymethylcellulose, benzalkonium chloride, water	No irritation effect *in vitro* and *in vivo* Sustained drug release *in vitro* Higher corneal residence time than marketed formulation *in vivo* 2–3-fold higher concentration in aqueous humor than marketed formulation *in vivo* Greater capacity in decreasing OII than marketed formulation *in vivo*	[189]
Loteprednol Etabonate	Nanoemulsion in-situ forming gel	Propylene glycol monocaprylate, poloxamer 407, poloxamer 188, benzalkonium chloride, artificial tear fluid, acetate buffer, cetalkonium chloride, glycerin, water	Zero-order drug release kinetics No irritation *in vitro* High entrapment efficiency (>95%) 2.54-fold higher bioavailability compared to marketed formulation *in vivo*	[190]
Prednisolone Acetate	Nanoparticles in gel	Acetic acid, PVA, sodium deoxycholate, methylparaben, propylparaben, hydroxypropylmethylcellulose, water	Entrapment efficiency (35 < X < 60%) Sustained drug release *in vitro* Greater anti-inflammatory effects *in vivo* than marketed formulation	[191]

PLGA: poly (lactic-*co*-glycolic acid), PVA: polyvinyl alcohol.

**Table 12 pharmaceutics-12-00570-t012:** Requirements of the particle size in ophthalmic preparations according to <798> US Pharmacopeia (USP) and 10th European Pharmacopeia (EP).

Maximal Number of Particles	Diameter			
	≥10 µm	≥25 µm	≥50 µm	≥90 µm
According to <798> USP	50 per mL	5 per mL	2 per mL	
According to EP 10^th^		20 per 10 µg of solid active substance	2 per 10 µg of solid active substance	None per 10 µg of solid active substance

**Table 13 pharmaceutics-12-00570-t013:** Model of the STE method inspired from OECD guidelines [204].

Cell Viability		UN GHS Classification	Applicability
At 5%	At 0.05%		
>70%	>70%	No category	No serious damage nor eye irritation effect
≤70%	>70%	No prediction can be made	No prediction can be made, eventual eye irritation
≤70%	≤70%	Category I	Serious eye damage

**Table 14 pharmaceutics-12-00570-t014:** Advantages and disadvantages of each type of cell.

Items	Primary Cell Cultures	Immortalized Cell Line	Reconstructed Tissue Culture
**Obtention**	From rabbit’s corneal tissue or human corneal epithelial cells by excising the tissue and allowing it to adhere	By maintaining the harvested cells in suitable growth medium and transfecting them with a viral vector to induce cell division	From bovine or human corneal tissue construct
**Advantages**	Relatively cheap and easy	Good correlation with excised rabbit cornea	Morphology similar to excised corneaMore accurate way to mimic the cornea
**Disadvantages**	Are not a true representation of the whole cornea	Exhibit abnormal gene expression and/or biological function

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
