# Peer review of "Recent Advances in the Design of Topical Ophthalmic Delivery Systems in the Treatment of Ocular Surface Inflammation and Their Biopharmaceutical Evaluation"

_pharmaceutics, 2020, doi:10.3390/pharmaceutics12060570_

Round 1

Reviewer 1 Report

pharmaceutics-824088
It is a well-written manuscript reviewing the available ocular anti-inflammatory options and alternative options being explored in the literature. Here are some comments.
• Since there are several comprehensive reviews already published focusing on ocular delivery (see example articles below covering anterior segment diseases/ocular inflammation), it would be helpful to the readers if the authors could cite previously published reviews in the manuscript and provide a brief explanation of how this review is different (the uniqueness of this manuscript) and would help the readers. o Lobo, Ann-Marie, Lucia Sobrin, and George N. Papaliodis. "Drug delivery options for the treatment of ocular inflammation." Seminars in ophthalmology. Vol. 25. No. 5-6. Taylor & Francis, 2010. o Lalu, Lida, et al. "Novel nanosystems for the treatment of ocular inflammation: current paradigms and future research directions." Journal of Controlled Release 268 (2017): 19-39. o Ahuja, Munish, et al. "Topical ocular delivery of NSAIDs." The AAPS journal 10.2 (2008): 229. o Janagam, Dileep R., Linfeng Wu, and Tao L. Lowe. "Nanoparticles for drug delivery to the anterior segment of the eye." Advanced drug delivery reviews 122 (2017): 31-64. o Cholkar, Kishore, et al. "Novel strategies for anterior segment ocular drug delivery." Journal of ocular pharmacology and therapeutics 29.2 (2013): 106-123.
• Sec 1. Para 3: Abbreviation API is slightly confusing, as per general pharmaceutical terminology API stands for ‘Active Pharmaceutical Ingredient’ and not ‘Active Principle Ingredient’, it may be helpful to change it to the regular definition.
• Sec 2.1. Table 1: Does this table present only the drugs used/prescribed for ocular inflammation irrespective of the route of administration (ROA) or does the list contain all the anti-inflammatory drugs approved (for all inflammation conditions), if so it may be helpful to limit the table to list the drugs that are only prescribed for treating ocular inflammation (all ROA) (Say if there are any drugs prescribed for ocular inflammation but taken through different ROA and not topical - they can be included but not all drugs.)
• Sec 3.1. Authors have put a nice effort in providing the total commercial drugs available “On February 26, 2019, we have identified 93 commercial drugs on the USA, European and French markets. Among these specialties, 35 contain an NSAID as API, 23 contain SAID, 30 correspond to anti-inflammatory API associated with anti-infective drugs (1 association with NSAID and 29 associations with SAID) and 5 contain CsA.”
o It looks like this list includes brands and generics, if so could you please mention it in the text?
o Also, it would be easy to see the categories visually - it would be helpful to the readers if authors could come-up with Pie charts showing the relative sizes of distribution/% of API categories (NSAID vs SAID, etc)
o Similarly, it would be helpful to create a Pie chart for different dosage forms as well (Solutions vs. Ointments vs. Suspensions, etc.)
• Section 3.2 Original Formulations – What does original formulation mean? Please define?
• Section 4 breakdown is not clear. “Examinations which have to be performed to determine the properties of each formulation may be divided into four parts: physicochemical characterization, biocompatibility evaluation, pharmacokinetics studies, and efficacy regarding the anti-inflammatory effect.” Then talked about “4.1. Sterility Assay”. Section 4.1 can be part of Section 4.2.
• “Section 4.1 Sterility Assay” it’s not called an “Assay” it is yes (Sterile) or no (non-sterile) test. Also, please cite other regulatory guidance documents as well (Ex. USP <71> STERILITY TESTS, etc) that provide testing guidelines/recommendations.
• Table 8: It would be helpful to be specific and mention the chapter (USP/EP) or section of the compendial/regulatory guidance documents.
• Although pharmacokinetic and in vivo safety and efficacy testing are needed during the development stage, they are dependant on the physicochemical properties of the API and finished product, therefore the in vivo testing (PK/Safety/Efficacy) can be completely separated from physical-chemical attributes (since these attributes need to be tested for showing batch to batch consistency on a routine bassis for releasing the lots to market). PK/Efficacy or other biotests may be separated from other attributes that are tested for meeting the batch regulatory specifications.

Author Response

Subject: comments for reviewers

Manuscript ID: pharmaceutics-824088

Title: Recent advances in the design of topical ophthalmic delivery system in the treatment of ocular surface inflammation and their biopharmaceutical evaluation

The authors thank the reviewers for the pertinence of their remarks and for helping us to improve the quality of our manuscript.

Please find the comments and answers proposed for the manuscript.

Reviewers'comments: Reviewer 1

It is a well-written manuscript reviewing the available ocular anti-inflammatory options and alternative options being explored in the literature. Here are some comments.

  1. Since there are several comprehensive reviews already published focusing on ocular delivery (see example articles below covering anterior segment diseases/ocular inflammation), it would be helpful to the readers if the authors could cite previously published reviews in the manuscript and provide a brief explanation of how this review is different (the uniqueness of this manuscript) and would help the readers.
    1. Lobo, Ann-Marie, Lucia Sobrin, and George N. Papaliodis. "Drug delivery options for the treatment of ocular inflammation." Seminars in ophthalmology. Vol. 25. No. 5-6. Taylor & Francis, 2010.
    2. Lalu, Lida, et al. "Novel nanosystems for the treatment of ocular inflammation: current paradigms and future research directions." Journal of Controlled Release 268 (2017): 19-39.
    3. Ahuja, Munish, et al. "Topical ocular delivery of NSAIDs." The AAPS journal 10.2 (2008): 229.
    4. Janagam, Dileep R., Linfeng Wu, and Tao L. Lowe. "Nanoparticles for drug delivery to the anterior segment of the eye." Advanced drug delivery reviews 122 (2017): 31-64.
    5. Cholkar, Kishore, et al. "Novel strategies for anterior segment ocular drug delivery." Journal of ocular pharmacology and therapeutics 29.2 (2013): 106-123.

We add this paragraph in the introduction in order to provide the uniqueness of this work : “Various studies have been published in this area, covering the administration of topical ocular anti-inflammatory drugs based on NSAIDs [1] and SAIDs [28] to the anterior and posterior segments of the eye, respectively.. The reviews of Janagam et al. [29], Lalu et al. [30] and Cholkar et al. [31] focused on the development of novel nanosystems for drugs from various pharmacological classes. The present review gives an updated insight of topical ophthalmic administration of SAIDs, NSAIDs and immunosuppressive agents in order to control the ocular inflammation. Indeed, immunosuppressive agents are specifically used in the treatment of inflammation associated with dry eye syndrome. Also, the review provides an exhaustive information of marketed specialties of French, European and US markets, brands and generics, specifying their indications and their complete formulation. In addition, conventional formulations and innovative ocular drug delivery systems are discussed. As well, the different characterization tools for biopharmaceutical evaluations of the systems are considered.”

  1. Sec 1. Para 3: Abbreviation API is slightly confusing, as per general pharmaceutical terminology API stands for ‘Active Pharmaceutical Ingredient’ and not ‘Active Principle Ingredient’, it may be helpful to change it to the regular definition.

The modification is made in the revised version of the review.

  1. Sec 2.1. Table 1: Does this table present only the drugs used/prescribed for ocular inflammation irrespective of the route of administration (ROA) or does the list contain all the anti-inflammatory 2 drugs approved (for all inflammation conditions), if so it may be helpful to limit the table to list the drugs that are only prescribed for treating ocular inflammation (all ROA) (Say if there are any drugs prescribed for ocular inflammation but taken through different ROA and not topical -they can be included but not all drugs.)

Table 1 contains all approved anti-inflammatory medications for all inflammatory conditions. It shows the few number of drugs formulated for the topical ophthalmic route. To support this fact, we add a sentence to the text: “As shown in Table 1, the ophthalmic topical route is largely under-endowed in anti-inflammatory drug specialties compared to other routes of administration.”

  1. Sec 3.1. Authors have put a nice effort in providing the total commercial drugs available “On February 26, 2019, we have identified 93 commercial drugs on the USA, European and French markets. Among these specialties, 35 contain an NSAID as API, 23 contain SAID, 30 correspond to anti-inflammatory API associated with anti-infective drugs (1 association with NSAID and 29 associations with SAID) and 5 contain CsA.”
    1. It looks like this list includes brands and generics, if so could you please mention it in the text?

This information was added in the revised text: “Among the topical dosage forms for ophthalmic drug delivery, eye drops solutions are quite popular since they are relatively well tolerated by patients, simple to prepare, to filter and to sterilize. On February 26, 2019, we have identified 93 commercial drugs, brands and generics, on the USA, European and French markets. “

  1. Also, it would be easy to see the categories visually-it would be helpful to the readers if authors could come-up with Pie charts showing the relative sizes of distribution/% of API categories (NSAID vs SAID, etc)

We have added figure 3 in the text:

Figure 3. Repartition of NSAID, SAID, CsA, NSAID + anti-infective drugs and SAID + anti-infective drugs on the USA, European and French markets

  1. Similarly, it would be helpful to create a Pie chart for different dosage forms as well (Solutions vs. Ointments vs. Suspensions, etc.)

Figure 4 was added in the revised version of the manuscript : “Among these 93 topical ocular specialties, 42 are formulated as solutions, 28 as suspensions, 15 as ointments, 5 as emulsions, 2 as gels and one as an intracanalicular insert (figure 4). “

Figure 4. Distribution of the different formulations of NSAID, SAID, CsA, NSAID + anti-infective drugs and SAID + anti-infective drugs on the USA, European and French markets

  1. Section 3.2 Original Formulations –What does original formulation mean? Please define?

This information was added in the revised text: “From conventional formulations, new formulations have been developed to increase residence time, decrease the frequency of instillation and finally increase the bioavailability of ophthalmic dosage forms [73]. These formulations innovate by the materials used, by the use of micro or nanoparticles or by the use of combined strategies.”

  1. Section 4 breakdown is not clear. “Examinations which have to be performed to determine the properties of each formulation may be divided into four parts: physicochemical characterization, biocompatibility evaluation, pharmacokinetics studies, and efficacy regarding the anti-inflammatory effect.” Then talked about “4.1. Sterility Assay”. Section 4.1 can be part of Section 4.2.

The modification is made in the revised version of the article.

  1. “Section 4.1 Sterility Assay” it’s not called an “Assay” it is yes (Sterile) or no (non-sterile) test. Also, please cite other regulatory guidance documents as well (Ex. USP <71> STERILITY TESTS, etc) that provide testing guidelines/recommendations.

The information is made in the revised version of the review article: “Sterility is one of the essential requirements for drug dosage forms applied on the eye ball. The sterility assay is well described in the 2.6.1 monograph of the European Pharmacopoeia or USP <71> sterility tests. It involves inoculation in aseptic conditions of the sample examined on two microbiological media”.

  1. Table 8: It would be helpful to be specific and mention the chapter (USP/EP) or section of the compendial/regulatory guidance documents.

As suggested by the reviewer, the chapter (USP/EP) was mentioned in the revised version of the review.

  1. Although pharmacokinetic and in vivo safety and efficacy testing are needed during the development stage, they are dependant on the physicochemical properties of the API and finished product, therefore the in vivo testing (PK/Safety/Efficacy) can be completely separated from physical-chemical attributes (since these attributes need to be tested for showing batch to batch consistency on a routine bassis for releasing the lots to market). PK/Efficacy or other biotests may be separated from other attributes that are tested for meeting the batch regulatory specifications

We thank reviewer 1 for his remark. As suggested, the paragraphs in Section 4 have been reorganized as follows to clearly distinguish between PK/safety/efficacy testing and testing that is performed to meet regulatory batch specifications.

“4.1 Sterility tests and physicochemical attributes

4.2 Biological evaluations

    4.2.1Toxicity and biocompatibility tests

    4.2.2 Pharmacokinetic studies

    4.2.3 Efficacy testing”

Similarly, the introduction to this chapter has been clarified along these lines.

“The examinations that must be performed to determine the properties of each formulation can be divided into two main categories: sterility testing and physicochemical characterization, and biological evaluations.”

Reviewer 2 Report

The manuscript entitled "Recent advances in the design of topical ophthalmic delivery system in the treatment of ocular surface inflammation and their biopharmaceutical evaluation" describes the types of drugs, formulations, and assay methods in the ophthalmic delivery system. I find the manuscript pretty well-written, and interesting. Also, it includes extensive cases of the recent progress that I would feel that the manuscript may be too lengthy. Therefore, I would recommend acceptance without further modification. 

Author Response

Subject: comments for reviewers

Manuscript ID: pharmaceutics-824088

Title: Recent advances in the design of topical ophthalmic delivery system in the treatment of ocular surface inflammation and their biopharmaceutical evaluation

The authors thank the reviewers for the pertinence of their remarks and for helping us to improve the quality of our manuscript.

Please find the comments and answers proposed for the manuscript.

Reviewers'comments: Reviewer 2

The manuscript entitled "Recent advances in the design of topical ophthalmic delivery system in the treatment of ocular surface inflammation and their biopharmaceutical evaluation" describes the types of drugs, formulations, and assay methods in the ophthalmic delivery system. I find the manuscript pretty well-written, and interesting. Also, it includes extensive cases of the recent progress that I would feel that the manuscript may be too lengthy. Therefore, I would recommend acceptance without further modification.

We sincerely thank the reviewer 2 for his very positive comments on our work.

Reviewer 3 Report

The text is fluent and correct, but I would have expected some more insight into the innovative therapies of neurodegenerative diseases treated with biological drugs.

I didn't see any mention of corticosteroids effects on IOP. 

My suggestion is to create a specific paragraph titled "innovative approaches with biological drugs".

This class of drugs is of great interest in the scientific panorama and has been widely used in anti-inflammatory therapies.

Author Response

Subject: comments for reviewers

Manuscript ID: pharmaceutics-824088

Title: Recent advances in the design of topical ophthalmic delivery system in the treatment of ocular surface inflammation and their biopharmaceutical evaluation

The authors thank the reviewers for the pertinence of their remarks and for helping us to improve the quality of our manuscript.

Please find the comments and answers proposed for the manuscript.

Reviewers'comments:  Reviewer 3

  1. The text is fluent and correct, but I would have expected some more insight into the innovative therapies of neurodegenerative diseases treated with biological drugs.

Concerning biological drugs used to treat neurodegenerative diseases, we plan to carry out a specific review on biological medicinal products later.

  1. I didn't see any mention of corticosteroids effects on IOP.

We have added a section in the review on adverse reactions, specifically.

  • Side effects

There are many important ocular side effects of NSAIDS, SAIDS and immunosuppressive agents. Topical administration of NSAIDs is common, but this treatment has clinically significant side effects, including ulceration and corneal perforation [52]. The adverse effects associated with the use of corticosteroid eye drops are different. These include elevated intraocular pressure and induced glaucoma, cataract formation, delayed wound healing and increased susceptibility to infection [53]. Furthermore, the most common reported side effect of CsA is ocular burning, reported in 17% of patients and approximately 3% of patients stopping the medication as a result of this side effect [7].

  1. My suggestion is to create a specific paragraph titled "innovative approaches with biological drugs".

As suggested previously, we plan to conduct a specific review on the use of innovative approaches with biological drugs for ocular application.

Reviewer 4 Report

This literature review can be described as a comprehensive and descriptive work organised in a convenient way for the reader. The number of sources used to produce this publication is quite impressive. Still, there can be some additions, e.g. in the very first table (Table 1 on page 4) authors mention different routes of administration for Betamethasone except topical ophthalmic route. Whilst in the following clinical guidelines and tutorials, eye drops and ointment with Betamethasone are mentioned: 

1) Chapter 12, page 105 in "Inflammatory and Infectious Ocular Disorders" (2020) (springer.com/gp/book/9789811385452); 

2) Page 32 in "The Uveitis Atlas" (2020) (springer.com/gp/book/9788132224099);

3) Chapter 12, page 432 in "Kanski's Clinical Ophthalmology" (2019) (elsevier.com/_dynamic/product-display?isbn=978-0-7020-7711-1);

4) Chapter 5.3.1, page 52 in "Emergency, Acute and Rapid Access Ophthalmology" (2019) (springer.com/gp/book/9783319923680);

5) Chapter 5, table 5.7.6, page 360 in "Ophthalmology" (2018) (elsevier.com/books/ophthalmology/yanoff/978-0-323-52819-1);

6) Chapter 25, page 1102 in "Oxford Handbook of Ophthalmology" (2018) (oxfordmedicine.com/view/10.1093/med/9780198804550.001.0001/med-9780198804550);

7) Chapter 8.7, page 143 in "Intraocular Inflammation " (2016) (springer.com/gp/book/9783540753858);

8) Chapter 22, table 22.7, page 700 in "Oxford American Handbook of Ophthalmology" (2011) (global.oup.com/academic/product/oxford-american-handbook-of-ophthalmology-9780195393446?cc=gb&lang=en&);

9) Chapter 63, page 245 in "Encyclopedia of the Eye" (2010) (elsevier.com/books/encyclopedia-of-the-eye/besharse/978-0-12-374198-1) (sciencedirect.com/science/article/pii/B9780123742032000634);

10) Volume 1, chapter 92, section 8, page 1145 in "Albert & Jakobiec's Principles & Practice of Ophthalmology" (2008) (elsevier.com/books/albert-and-jakobiecs-principles-and-practice-of-ophthalmology/albert/978-1-4160-0016-7);

11) Page 209 in "Ophthalmic Drugs" (2006) (elsevier.com/books/ophthalmic-drugs/9780750688642).

In the end, just a few more things. There is a typing mistake on page 9 missing the letter "n" in the name of "Alcon Laboratories". Another typing mistake appears on page 20 of the manuscript (crossed number 15). In the tables 4-7 (pages 11-19) some of the empty boxes of the excipients include hyphen while others do not. Also, as a suggestion, the name of this manuscript can be slightly shortened. 

In general, this literature review is well organised and written. It requires minor English language spell checks and is recommended for publication after minor corrections. 

Author Response

Subject: comments for reviewers

Manuscript ID: pharmaceutics-824088

Title: Recent advances in the design of topical ophthalmic delivery system in the treatment of ocular surface inflammation and their biopharmaceutical evaluation

The authors thank the reviewers for the pertinence of their remarks and for helping us to improve the quality of our manuscript.

Please find the comments and answers proposed for the manuscript.

Reviewers'comments:  Reviewer 4

This literature review can be described as a comprehensive and descriptive work organised in a convenient way for the reader. The number of sources used to produce this publication is quite impressive.

  1. Still, there can be some additions, e.g. in the very first table (Table 1 on page 4) authors mention different routes of administration for Betamethasone except topical ophthalmic route. Whilst in the following clinical guidelines and tutorials, eye drops and ointment with Betamethasone are mentioned: 
    1. Chapter 12, page 105 in "Inflammatory and Infectious Ocular Disorders" (2020) (springer.com/gp/book/9789811385452); 
    2. Page 32 in "The Uveitis Atlas" (2020) (springer.com/gp/book/9788132224099);
    3. Chapter 12, page 432 in "Kanski's Clinical Ophthalmology" (2019) (elsevier.com/_dynamic/product-display?isbn=978-0-7020-7711-1);
    4. Chapter 5.3.1, page 52 in "Emergency, Acute and Rapid Access Ophthalmology" (2019) (springer.com/gp/book/9783319923680);
    5. Chapter 5, table 5.7.6, page 360 in "Ophthalmology" (2018) (elsevier.com/books/ophthalmology/yanoff/978-0-323-52819-1);
    6. Chapter 25, page 1102 in "Oxford Handbook of Ophthalmology" (2018) (oxfordmedicine.com/view/10.1093/med/9780198804550.001.0001/med-9780198804550);
    7. Chapter 8.7, page 143 in "Intraocular Inflammation " (2016) (springer.com/gp/book/9783540753858);
    8. Chapter 22, table 22.7, page 700 in "Oxford American Handbook of Ophthalmology" (2011) (global.oup.com/academic/product/oxford-american-handbook-of-ophthalmology-9780195393446?cc=gb&lang=en&);
    9. Chapter 63, page 245 in "Encyclopedia of the Eye" (2010) (elsevier.com/books/encyclopedia-of-the-eye/besharse/978-0-12-374198-1) (sciencedirect.com/science/article/pii/B9780123742032000634);
    10. Volume 1, chapter 92, section 8, page 1145 in "Albert & Jakobiec's Principles & Practice of Ophthalmology" (2008) (elsevier.com/books/albert-and-jakobiecs-principles-and-practice-of-ophthalmology/albert/978-1-4160-0016-7);
    11. Page 209 in "Ophthalmic Drugs" (2006) (elsevier.com/books/ophthalmic-drugs/9780750688642).

As suggested by the reviewer 4, the table 1 was modified in the revised version of the review.

  1. In the end, just a few more things. There is a typing mistake on page 9 missing the letter "n" in the name of "Alcon Laboratories".

The modification is made in the revised version of the article.

  1. Another typing mistake appears on page 20 of the manuscript (crossed number 15).

The modification is made in the revised version of the article.

  1. In the tables 4-7 (pages 11-19) some of the empty boxes of the excipients include hyphen while others do not. Also, as a suggestion, the name of this manuscript can be slightly shortened. 

The modifications in the tables 4-7 are made in the revised version of the article.

    5. In general, this literature review is well organised and written. It requires minor English language spell checks and is recommended for publication after minor corrections. 

We sincerely thank the reviewer 4 for his positive comments on our work.